# Impact of socioeconomic- and lifestyle-related risk factors on poor mental health conditions: A nationwide longitudinal 5-wave panel study in Japan

Miwako Nagasu[1]*, Isamu Yamamoto[2]

1 Faculty of Economics, Keio University, Tokyo, Japan, 2 Faculty of Business and Commerce, Keio University, Tokyo, Japan

* mnagasu555@gmail.com

## Abstract

The association of socioeconomic status and lifestyle behaviours on mental health appears well-established in the literature, as several studies report that better socioeconomic status such as higher levels of disposable income and employment as well as practising healthy lifestyles can enhance mental well-being. However, the reliance on cross-sectional correlations and lack of adequate statistical controls are possible limitations. This study aims to add the evidence of longitudinal association to the literature by using Japanese representative longitudinal household panel data. We employed panel data analytical techniques such as the random-effects conditional logistic regression (RE-CLR) and the fixed-effects conditional logistic regression (FE-CLR) models with possible time variant confounders being controlled. Our sample was comprised of 14,717 observations of 3,501 individuals aged 22–59 years for five waves of the Japanese Household Panel Survey. We confirmed many of the factors associated with mental health reported in existing studies by analysing cross-sectional data. These significant associations are also longitudinal (within) associations estimated by the FE-CLR models. Such factors include unemployment, low household income, short nightly sleeping duration, and lack of exercise. However, we also found that several factors such as disposable income, living alone, and drinking habits are not significantly associated with mental health in the FE-CRL models. The results imply the reverse causality that poor mental health conditions cause lower disposal income, possibly due to the inability to exhibit higher productivity, but an increase in disposal income would not necessarily improve mental health conditions. In this case, aggressive policy interventions to increase the disposal income of people of lower socioeconomic backgrounds would not necessarily be effective to minimize health inequalities.

**Data Availability Statement:** The datasets used and analysed in the current study are available on the website of the Panel Data Research Center at Keio University: https://www.pdrc.keio.ac.jp/en/. The website explains how to register the request

form. The centre may approve within a few days on reasonable request.

**Funding:** This study was supported by JSPS KAKENHI Grant Number JP 17H06086 and 18K01659. However, the funders had no role in the design of the study, in collection, analysis, and interpretation of data, or in writing the manuscript.

**Competing interests:** The authors have declared that no competing interests exist.

**Abbreviations:** WHO, World Health Organisation; 95% CI, 95% confidence interval; AOR, Adjusted odds ratio; GHQ-12, General Health Questionnaire 12-items; JHPS/KHPS, the Japan Household Panel Survey; RE-CLR, Random-effects conditional logistic regression model; FE-CLR, Fixed-effects conditional logistic regression model; SES, Socioeconomic status.

# Introduction

Health inequalities have been receiving significant attention in many countries [1, 2]. The World Health Organisation has recently appealed to governments worldwide with this phrase, 'Health inequalities are unfair,' which served as a worldwide call to action to minimise health inequalities through governmental policies [3]. Subsequent studies have found that differences in socioeconomic status (SES) are one of the causes of health inequalities and are associated with health outcomes [2, 4, 5]. These studies usually examined participants' income, educational levels, occupation and employment status which are the socioeconomic factors that have been proven to be main determinants related to individuals' mental and physical health [1, 6–8]https://www.ncbi.nlm.nih.gov/pmc/articles/PMC3783534/ - R36.

Mental illness has become a notable public health concern on a global scale, and has been directly related to physical disorders and suicides among those affected [9]. A previous study reported that mental illness and SES such as low income and unemployment were significantly associated with higher risks for committing suicide [5]. Japan was shown to be one of the countries with the highest suicide rates in the world [10, 11]; thus, it is important to investigate any possible associations between suicide and mental health, SES, and lifestyle-related factors [12].

Cross-sectional studies have shown that healthy lifestyle factors are positively associated with better mental health outcomes [13]. In that regard, the relationship between mental health conditions and lifestyle factors such as sleeping duration [14], habitual physical exercise [15], smoking habits [16, 17], and alcohol consumption [18] have been studied to find methods of preventing mental illness. For example, Golzier et al. analysed a cohort study and reported that shorter sleep duration is linearly associated with psychological distress [14]. The cross-sectional study and its follow-up study reported that habitual physical exercise was associated with better health outcomes [15, 19]. Additional cross-sectional studies found that smoking and alcohol problems are positively associated with depressive symptoms [16, 18]. These studies suggest that increasing healthy lifestyle practises may promote better health outcomes. However, the main limitation of these findings is that confounding remains possible when the study design was cross-sectional, unless all studies adjusted variables.

Deepening this discussion, there appear to be connections among lifestyle practises, mental disorders, SES, and poorer health conditions [20]. In particular, mental disorders are frequently associated with people with lower SES [5, 21]. Some studies report that people with low SES are more likely to practise unhealthy behaviours than those with high SES [4, 22]. For example, people with lower SES are more likely to be smokers, so they suffer from the effects of smoking [23]. This may help explain the background of income-related health inequality, and so it is important to determine causal links between poor health conditions and low levels of SES. Furthermore, it has been essential to discuss the probable causality running in both directions: poor SES may promote poor mental health conditions, and vice versa [24].

Therefore, instead of examining cross-sectional data, we analysed the panel (longitudinal) dataset by using random-effects and fixed-effects longitudinal regression models to avoid the potential bias from unobserved confounders. The random-effects models do not control for unobservable and time-invariant individual attributes, so the estimated results of these models should resemble those of cross-sectional studies. However, the fixed-effects models are able to control for unobservable and person-specific time-invariant heterogeneity and reverse causality caused by time-invariant factors, so the estimated results of these models may not resemble those of cross-sectional studies [25]. For example, some people may be resilient and cope with stress innately, but some may not. If such an unobservable heterogeneity factor is correlated with explanatory variables such as SES and lifestyle-related factors, the estimates from a cross-

sectional regression or random-effects model would become inconsistent due to the omitted variable bias. Furthermore, in such cases, it often becomes impossible to interpret the causal relationships from the estimated results. However, the use of fixed-effects models often allows for the interpretation of the estimated results as reflections of a causal relationship as far as the time-invariant unobservable heterogeneity factor brings about reverse causality. Notwithstanding, it should be noted that, when performing this type of analysis, it is also important to use instrument variables to address the reverse causality caused by time-variant factors. However, it is not easy to find appropriate instrument variables that are dependent on explanatory variables but independent of dependent variables. We have not been able to find appropriate instrument variables for this study, but we suggest that future research should seek to use them to overcome the possible reverse causality related to time-variant factors.

To summarize, we utilised fixed-effects models to control time-invariant factors, so as to help in the identification of causal relationships and to eliminate possible biases, thereby leading to more consistent estimated results by analysing longitudinal panel data [26]. This study aims to build on the cross-sectional research cited which suggest positive associations between SES and lifestyle-related factors on participants' mental health conditions, even after controlling for unobservable person-specific time-invariant individual attributes in the fixed-effects model. The objectives of this study are [1] to examine the prevalence of poor mental health conditions among Japanese individuals aged 22 to 59 years and [2] to analyse the differences between the estimates from random- and fixed-effects models, so as to help identify the SES and lifestyle-related risk factors for poor mental health conditions. Our findings may provide further evidence that incentivising people to practise healthier lifestyles may benefit mental health.

## Methods

For this study, we used longitudinal panel data from the Japan Household Panel Survey (JHPS/KHPS). The JHPS/KHPS was developed by the Panel Data Research Centre at Keio University, Japan. This panel dataset is valuable because it has been providing nationally representative samples that allow for the study of Japanese individuals' conditions related to their socioeconomic status, income and poverty dynamics, disparities in health quality, and health behaviours for the past two decades.

### Sampling of the respondents

The JHPS/KHPS utilises a two-stage stratified random sampling method. According to the National Census Survey, Japan was stratified into 24 levels based on regional and city classifications. The number of samples for each level were calculated by using basic resident register population ratios with a range of 5 to 10 samples selected for each level.

The KHPS sample began with 4,005 respondents in 2004, and 1,400 and 1,000 respondents were added in 2007 and 2012, respectively. Subsequently, after being renamed JHPS, the sample began with 4,022 people in 2009. From 2004 to 2017, a total of 10,458 respondents were selected at random from the basic resident register system in Japan. It should be noted that the number of respondents gradually declined during the long study period. The JHPS/KHPS survey was carried out every year from February to March, and the questionnaire was distributed to respondents who had participated the previous year. This study utilised a 5-wave dataset that was collected from 2014 to 2018. In Japan, public pension benefits begin at the age of 60 to 65 years, so the socioeconomic status, such as disposable income, for this specific age group of the population is different from the other age groups. Thus, 10,185 observations of 2,204 individuals were excluded because the ages of the respondents were not between 22 and 59 years

**Table 1. The number of respondents and response rates for each survey and the number of recruited participants from each survey.**

| | JHPS | | | | | KHPS | | | | Total |
|---|---|---|---|---|---|---|---|---|---|---|
| Wave | Survey year | Number of respondents in total[1] | Response rate (%) | Respondents in this study | Wave | Survey year | Number of respondents in total | Response rate (%) | Respondents in this study | |
| 6 | 2014 | 2,358 | 91.1 | 1,385 | 11 | 2014 | 3,312 | 92.6 | 2,102 | 3,487 |
| 7 | 2015 | 2,198 | 93.0 | 1,253 | 12 | 2015 | 3,124 | 98.8 | 1,939 | 3,192 |
| 8 | 2016 | 2,048 | 92.8 | 1,144 | 13 | 2016 | 2,945 | 94.0 | 1,776 | 2,920 |
| 9 | 2017 | 1,885 | 91.9 | 1,057 | 14 | 2017 | 2,741 | 92.7 | 1,628 | 2,685 |
| 10 | 2018 | 1,742 | 92.2 | 970 | 15 | 2018 | 2,549 | 93.0 | 1,463 | 2,433 |
| Total | | 10,231 | 92.2 | **5,809** | Total | | 14,671 | 94.2 | **8,908** | **14,717** |

[1] Number of respondents in total: the number of collected questionnaires which were completed by the respondents.

during the corresponding years of their possible participation. In total, we used data from 14,717 observations of 3,501 individuals aged 22 to 59. Table 1 details the number of respondents and response rates for each survey and the number of recruited participants from each survey for this study. All respondents received an informed consent form about the aims and details of the study, which also provided information about the anonymity and confidentiality of the replies. Before receiving the questionnaire, those selected were asked to participate in the study, and after agreeing, the questionnaire was sent or brought to their home by research assistants. Signed written consent by the participants was obtained for the study.

## Study variables

A self-administered questionnaire collected participants' information related to socioeconomic factors, lifestyle-related factors, and mental health outcomes.

**Socioeconomic factors.** To assess participants' SES, via the questionnaire, they were asked about their gender, age, employment status, the number of persons in the household (living status), and disposable income per household. Participants included only those aged 22 to 59 years at the time of participation. Employment status was divided into four groups: unemployed, self-employed, regular employee, and non-regular employee. The number of persons in the household was categorised into two groups: living with someone ($\geq$ 2 people) and living alone. All participants were asked for their annual household disposable income for the year prior to participation, which included disposable incomes of all household members excluding tax and social insurance fees. The disposable income per household was divided into three groups: low level ($<$ 2,000K yen), middle level (2,000K–6,000K yen), and high level ($\geq$ 6,000K yen).

**Lifestyle factors.** To assess participants' lifestyle practices, via the questionnaire, they were asked about their sleep duration during the week, physical exercise frequency, and smoking and drinking alcohol habits. For sleep duration, the question used was 'How many hours do you usually sleep each weekday night?' The responses were categorised into three groups: $\geq$ 7 hours, 6–7 hours, and $<$ 6 hours. Regarding physical exercise frequency, the question used was: 'Excluding work-related activities, how many days per week do you perform physical exercise in which you sweat?' The answers were categorised into three groups: $\geq$ 3 days/week, $\leq$ 2 days/week, and no exercise. Regarding smoking habits, the question used was 'Do you smoke?' Participants were categorised as never-smokers, ex-smokers, and current smokers who smoked sometimes/every day. Drinking habits were assessed by the question: 'How often do you drink alcohol?' The answers were categorised into the following groups: Never, $\leq$ 2 times/week, $\geq$ 3 times/week.

**Health outcomes.** Participants' mental health status was measured by the General Health Questionnaire 12-items (GHQ-12) [27, 28], written in the Japanese language. This questionnaire served as a screening measure to detect nonpsychotic psychiatric diseases, and it was comprised of 12 questions about participants' feelings over the previous few weeks. The questions included the following: Have you recently (1) been able to concentrate on whatever you're doing, (2) lost much sleep over worry, (3) felt that you were playing a useful part in general, (4) felt capable of making decisions, (5) felt constantly under strain, (6) felt you couldn't overcome your difficulties, (7) been able to enjoy your normal day-to-day activities, (8) been able to face up to problems, (9) been feeling unhappy or depressed, (10) been losing confidence in yourself, (11) been thinking of yourself as a worthless person, (12) been feeling reasonably happy, all things considered. In order to assess the severity of participants' psychological distress, we utilised a scoring system described herein: the response categories (1, 2, 3, and 4) were converted into corresponding binary values (0, 0, 1, and 1) to calculate the total score of the 12 questions. The subjects were then divided into two groups: those with higher scores/ poor mental health conditions: $\geq 4$ points; and those with lower scores/good mental health conditions: $\leq 3$ points [27].

## Statistical methods

Participants' demographic characteristics were analysed through mean and standard deviations values and percentages. Further, we analysed the association between socioeconomic and lifestyle factors by gender through adjusted prevalence odds ratios, and utilised 95% confidence intervals (95% CI) of participants' scores (0: $\leq 3$ points, 1: $\geq 4$ points) from GHQ-12 by using both the random-effects conditional logistic regression (RE-CLR) and the fixed-effects conditional logistic regression (FE-CLR) models. These models are commonly used to analyse panel data [29]. Further, applying these two methods can help eliminate bias and improve consistency of the results [29], as well as help estimate the longitudinal association between socioeconomic- and lifestyle-related factors on participants' mental health outcomes. We further evaluated the models by using the Hausman specification test to identify the better model for each analysis [29]. We controlled the results for all factors: age, number of persons in the household, employment status, annual disposable income per household, and lifestyle-related factors. Data was analysed separately by gender because some studies reported that the prevalence of mental health conditions and lifestyle factors were different between men and women [30, 31]. The statistical analysis was completed by using the SPSS 25.0 and STATA MP 15 computer package.

The Institutional Review Board, Institute for Economic Studies, Keio University approved this study (Reference number 15002).

## Results

This study used a 5-wave panel dataset and analysed data for a total of 14,717 participants including 7,215 men (49.0%) and 7,502 women (51.0%). Participants' descriptive statistics are detailed in Table 2.

In terms of participants' mental health conditions, 36.1% of men and 42.0% of women were shown to have poor mental health conditions ($\geq 4$ GHQ-12 score). The results indicated statistically significant differences between men and women for every year and pooled data of all waves. Fig 1 shows participants' GHQ-12 scores by gender.

The results of the estimated associations between the GHQ scores and risk factors based on the random-effects conditional logistic regression model (RE-CLR), the fixed-effects conditional logistic regression (FE-CLR) model, and the results of the Hausman tests are shown in Table 3.

**Table 2. Participants' demographic characteristics.**

| Variables | Group | Total | | Men | | Women | |
|---|---|---|---|---|---|---|---|
| | | N/mean | %/SD | N/mean | %/SD | N/mean | %/SD |
| Wave | | | | | | | |
| | Men | 7,215 | 49.0% | | | | |
| | Women | 7,502 | 51.0% | | | | |
| Age (in years) | | 45.3 | 8.7 | 45.2 | 8.7 | 45.4 | 8.7 |
| Number of persons in the household | | | | | | | |
| | ≥ 2 people | 13,487 | 92.1% | 6,415 | 89.4% | 7,072 | 94.7% |
| | One person | 1,155 | 7.9% | 757 | 10.6% | 398 | 5.3% |
| Employment status | | | | | | | |
| | Unemployed | 1,885 | 12.9% | 271 | 3.8% | 1,614 | 21.7% |
| | Self-employed | 1,818 | 12.4% | 1,106 | 15.4% | 712 | 9.6% |
| | Regular employee | 7,126 | 48.7% | 5,332 | 74.3% | 1,794 | 24.1% |
| | Non-regular employee | 3,796 | 26.0% | 466 | 6.5% | 3,330 | 44.7% |
| Disposable income per household | | | | | | | |
| | ≥ 6,000K | 4,856 | 36.9% | 2,424 | 37.2% | 2,432 | 36.6% |
| | 2,000K-< 6,000K | 7,373 | 56.0% | 3,720 | 57.1% | 3,653 | 54.9% |
| | < 2,000K | 935 | 7.1% | 370 | 5.7% | 565 | 8.5% |
| | in Japanese Yen | 542.8 | 311.3 | 542.2 | 295.9 | 543.3 | 325.6 |
| Sleep duration. weekdays | | | | | | | |
| | ≥ 7 hours | 5,450 | 37.6% | 2,710 | 38.2% | 2,740 | 37.0% |
| | 6–7 hours | 5,595 | 38.6% | 2,769 | 39.1% | 2,826 | 38.2% |
| | < 6 hours | 3,448 | 23.8% | 1,607 | 22.7% | 1,841 | 24.9% |
| Physical exercise | | | | | | | |
| | ≥ 3 days/week | 1,415 | 9.7% | 729 | 10.2% | 686 | 9.2% |
| | ≤ 2 days/week | 2,637 | 18.0% | 1,518 | 21.2% | 1,119 | 15.0% |
| | No exercise | 10,568 | 72.3% | 4,914 | 68.6% | 5,654 | 75.8% |
| Smoking habit | | | | | | | |
| | Never | 7,888 | 53.7% | 2,524 | 35.0% | 5,364 | 71.6% |
| | Quit | 3,423 | 23.3% | 2,220 | 30.8% | 1,203 | 16.1% |
| | Sometimes + everyday | 3,379 | 23.0% | 2,459 | 34.1% | 920 | 12.3% |
| Drinking alcohol habit | | | | | | | |
| | Never | 5,192 | 35.4% | 1,746 | 24.3% | 3,446 | 46.2% |
| | ≤ 2 times/week | 5,130 | 35.0% | 2,415 | 33.6% | 2,715 | 36.4% |
| | ≥ 3 times/week | 4,328 | 29.5% | 3,027 | 42.1% | 1,301 | 17.4% |
| GHQ score | | | | | | | |
| | ≥ 4 points (poor) | 5,713 | 39.1% | 2,581 | 36.1% | 3,132 | 42.0% |
| | ≤ 3 points | 8,891 | 60.9% | 4,562 | 63.9% | 4,329 | 58.0% |
| | 0 point—12 points | 3.4 | 3.4 | 3.2 | 3.4 | 3.6 | 3.4 |

The analysis made by computing adjusted odds ratios (AOR) among all participants and by gender indicated the differences among the potential risk factors. Among all participants, the Hausman test supported the FE-CLR model, and the estimated results indicated significant associations between poor mental health conditions and variables such as being unemployed (AOR 1.727 [95% CI: 1.134–2.629]), low level of disposable income per household (< 2,000K yen: AOR 1.441 [95% CI: 0.993–2.092]), having a short sleep duration (< 6 hours: AOR 1.503 [95% CI: 1.177–1.919]), and lack of physical exercise (AOR 1.418 [95% CI: 1.062–1.895]). It is important to note that the middle variables for disposable income per household (between

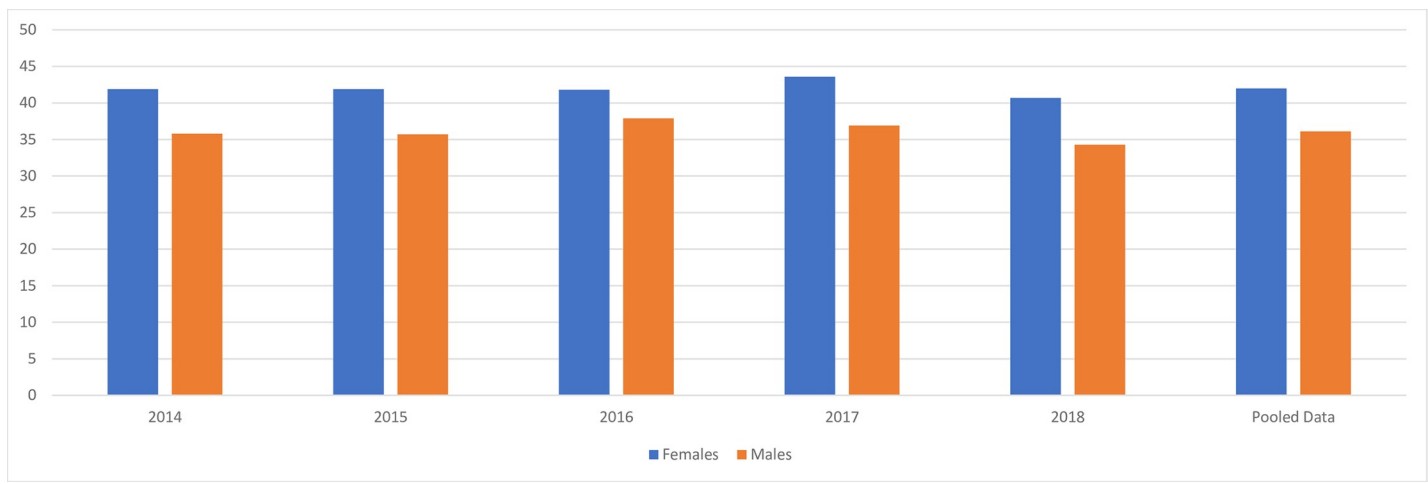

**Fig 1. Participants' General Health Questionnaire 12-item scores by gender.** The levels of the General Health Questionnaire 12-item scores: 0 = ≤ 3 points, 1 = ≥ 4 points. Results of GHQ scores by gender: 2014: ***p < .001, 2015: ***p < .001, 2016: * p < .05, 2017: ***p < .001, 2018: **p < .01, Pooled data: n.s.

2,000K–6,000K yen: AOR 1.254 [95% CI: 1.075–1.464]) and for drinking habits (≥ 3 times/week: AOR 0.785 [95% CI: 0.629–0.979]) exhibited significant effects on poor mental health only in the RE-CLR model.

Among male participants, the Hausman test supported the FE-CLR model, and the estimated results indicated significant associations between poor mental health conditions and variables such as being unemployed (AOR 5.852 [95% CI: 2.376–14.41]), having a short sleep duration (AOR 1.456 [95% CI: 1.012–2.095]), and lack of physical exercise (AOR 1.531 [95% CI: 1.013–2.315]). Once again, only in the RE-CLR model, variables such as living alone (AOR 1.422 [95% CI: 0.946–2.139]) and low level of disposable income (AOR 1.714 [95% CI: 1.082–2.731]) showed significant effects on poor mental health. However, these variables did not show significant associations in the FE-CLR model.

Among female participants, the RE-CLR model was supported by the Hausman test, and the results indicated significant associations between poor mental health conditions and middle (AOR 1.386 [95% CI: 1.126–1.705]) and low levels of household disposable income (AOR 1.980 [95% CI: 1.351–2.901]), having a middle (6–7 hours: AOR 1.277 [95% CI: 1.054–1.547]) and a short sleep duration (AOR 1.662 [95% CI: 1.283–2.151]), and lack of physical exercise (AOR 1.540 [95% CI: 1.099–2.160]). However, variables such as self-employment (AOR 0.608 [95% CI: 0.402–0.920]), being a non-regular employee (AOR 0.699 [95% CI: 0.527–0.928]), and drinking alcohol more than 3 times/week (AOR 0.752 [95% CI: 0.547–1.035]) were inversely associated with poor mental health conditions.

## Discussion

In this study, we utilised a 5-wave longitudinal panel survey to examine the prevalence of poor mental health conditions among men and women in Japan, which was measured by participants' GHQ-12 scores. Among the AOR analysis both in the RE-CLR and FE-CLR models, for all participants, the results indicated significant associations between poor mental health conditions and being unemployed, low levels of disposable income, short sleep durations, and lack of physical exercise. Among men, there were significant associations between poor mental health conditions and being unemployed, short sleep duration, and lack of physical exercise. Among women, results indicated significant associations between poor mental health conditions and low levels of household disposable income, short sleep duration, and lack of physical

**Table 3. Estimated associations between participants' General Health Questionnaire 12-item scores and risk factors by gender based on the random-effects conditional logistic regression models, on the fixed-effects conditional logistic regression models and on the Hausman tests.**

| | | All samples | | Men | | Women | |
|---|---|---|---|---|---|---|---|
| | | AOR(C.I.) [1] | AOR(C.I.)[2] | AOR(C.I.)[2] | AOR(C.I.)[2] | AOR(C.I.)[2] | AOR(C.I.)[2] |
| Sex | Men | Ref. | | | | | |
| | Women | 1.419*** | | | | | |
| | | (1.107–1.819) | | | | | |
| Age | Under 39 | Ref. | Ref. | Ref. | Ref. | Ref. | Ref. |
| | 40–49 | 0.971 | *1.117* | 0.857 | *1.121* | *1.146* | 1.120 |
| | | (0.791–1.190) | *(0.815–1.53[2])* | (0.623–1.179) | *(0.692–1.815)* | *(0.880–1.49[2])* | (0.738–1.700) |
| | 50–59 | 0.836 | *1.082* | 0.766 | *0.935* | *0.936* | 1.184 |
| | | (0.666–1.049) | *(0.693–1.69[1])* | (0.538–1.09[2]) | *(0.474–1.846)* | *(0.697–1.256)* | (0.653–2.149) |
| Number of persons in the household | ≥ 2 people | Ref. | Ref. | Ref. | Ref. | Ref. | Ref. |
| | One person | 1.286 | *0.999* | 1.422* | *1.085* | *0.994* | 0.905 |
| | | (0.946–1.749) | *(0.634–1.575)* | (0.946–2.139) | *(0.612–1.924)* | *(0.620–1.594)* | (0.430–1.904) |
| Employment status | unemployed | 1.501*** | *1.727** | 8.035*** | *5.852*** | *0.840* | 1.097 |
| | | (1.122–2.008) | *(1.134–2.629)* | (4.219–15.30) | *(2.376–14.4[1])* | *(0.599–1.180)* | (0.635–1.896) |
| | self-employed | 0.832 | *1.193* | 0.896 | *1.134* | *0.608** | 0.937 |
| | | (0.626–1.105) | *(0.727–1.958)* | (0.605–1.327) | *(0.563–2.286)* | *(0.402–0.920)* | (0.451–1.944) |
| | regular employee | Ref. | Ref. | Ref. | Ref. | Ref. | Ref. |
| | non-regular employee | 0.983 | *0.958* | 1.332 | *1.083* | *0.699** | 0.764 |
| | | (0.774–1.247) | *(0.680–1.350)* | (0.808–2.196) | *(0.565–2.077)* | *(0.527–0.928)* | (0.485–1.203) |
| Disposable income per household | ≥ 6,000K | Ref. | Ref. | Ref. | Ref. | Ref. | Ref. |
| | 2,000K-< 6,000K | 1.254*** | *1.118* | 1.150 | *0.970* | *1.386*** | 1.273* |
| | | (1.075–1.464) | *(0.924–1.35[2])* | (0.913–1.448) | *(0.728–1.293)* | *(1.126–1.705)* | (0.986–1.644) |
| | < 2,000K | 1.977*** | *1.441** | 1.714** | *1.382* | *1.980*** | 1.464 |
| | | (1.475–2.650) | *(0.993–2.09[2])* | (1.082–2.713) | *(0.782–2.440)* | *(1.351–2.90[1])* | (0.881–2.434) |
| Sleep duration of the respondent | ≥ 7 hours | Ref. | Ref. | Ref. | Ref. | Ref. | Ref. |
| | 6–7 hours | 1.100 | *1.128* | 0.939 | *0.988* | *1.277** | 1.285** |
| | | (0.951–1.27[2]) | *(0.952–1.335)* | (0.752–1.173) | *(0.761–1.28[2])* | *(1.054–1.547)* | (1.028–1.606) |
| | < 6 hours | 1.644*** | *1.503*** | 1.613*** | *1.456** | *1.662*** | 1.500** |
| | | (1.355–1.993) | *(1.177–1.919)* | (1.203–2.16[1]) | *(1.012–2.095)* | *(1.283–2.15[1])* | (1.070–2.104) |
| Physical exercise | ≥ 3 days/week | Ref. | Ref. | Ref. | Ref. | Ref. | Ref. |
| | ≤ 2 days/week | 1.175 | *1.066* | 1.128 | *0.979* | *1.265* | 1.184 |
| | | (0.897–1.538) | *(0.792–1.435)* | (0.768–1.656) | *(0.646–1.485)* | *(0.864–1.85[1])* | (0.767–1.828) |
| | No exercise | 1.637*** | *1.418** | 1.747*** | *1.531** | *1.540** | 1.326 |
| | | (1.284–2.088) | *(1.062–1.895)* | (1.227–2.488) | *(1.013–2.315)* | *(1.099–2.160)* | (0.874–2.01[2]) |
| Smoking habit | Never | Ref. | Ref. | Ref. | Ref. | Ref. | Ref. |
| | Quit | 1.135 | *0.929* | 1.119 | *1.054* | *1.176* | 0.856 |
| | | (0.909–1.419) | *(0.579–1.490)* | (0.794–1.578) | *(0.507–2.193)* | *(0.880–1.570)* | (0.476–1.540) |
| | Sometimes + everyday | 1.217 | *0.940* | 1.201 | *1.176* | *1.242* | 0.714 |
| | | (0.950–1.560) | *(0.520–1.700)* | (0.850–1.697) | *(0.503–2.750)* | *(0.854–1.805)* | (0.302–1.685) |
| Drinking alcohol habit | Never | | | | | | |
| | ≤ 2 times/week | 0.893 | *1.033* | 0.817 | *0.928* | *0.945* | 1.073 |
| | | (0.745–1.069) | *(0.809–1.320)* | (0.602–1.108) | *(0.609–1.413)* | *(0.758–1.179)* | (0.793–1.45[2]) |
| | ≥ 3 times/week | 0.785** | *0.775* | 0.783 | *0.679* | *0.752* | 0.819 |
| | | (0.629–0.979) | *(0.552–1.088)* | (0.564–1.087) | *(0.407–1.133)* | *(0.547–1.035)* | (0.503–1.334) |
| | Constant | 0.189*** | | 0.201*** | | *0.309*** | |
| | | (0.129–0.278) | | (0.116–0.350) | | *(0.188–0.506)* | |

*(Continued)*

**Table 3.** (Continued)

|  |  | All samples | | Men | | Women | |
| --- | --- | --- | --- | --- | --- | --- | --- |
|  |  | AOR(C.I.) [1) | AOR(C.I.) [2) | AOR(C.I.) [2) | AOR(C.I.) [2) | AOR(C.I.) [2) | AOR(C.I.) [2) |
|  | Observations | 12,681 | *5,992* | 6,270 | *2,748* | *6,411* | 3,244 |
|  | Number of respondents | 3,359 |  | 1,657 |  | *1,702* |  |
|  | Model Type | RE | *FE** | RE | *FE** | *RE** | FE |
|  | Hausman Test | 0.0061 | | 0.0586 | | 0.4554 | |

[1) Adjusted odds ratios (AOR) with 95% CI (adjusted for sex, age, number of persons in the household, employment status, disposable income per household, sleep duration in weekdays, physical exercise, smoking habit, and drinking alcohol).

[2) Adjusted odds ratios with 95% CI (adjusted for age, number of persons in the household, employment status, sleep duration in weekdays, physical exercise, smoking habit, and drinking alcohol).

Bold ratios: statistically significant results.

The levels of the General Health Questionnaire 12-item scores: 0 = ≤ 3 points, 1 = ≥ 4 points.

Robust cieform in parentheses.

Ref.: Reference [1].

*** p<0.01

** p<0.05

* p<0.1.

exercise. Contrastingly, self-employment, being a non-regular employee, and drinking alcohol more than 3 times per week were inversely associated with poor mental health conditions. In general, these results were consistent with the results obtained from previous cross-sectional studies that addressed the risk factors of poor mental health conditions, with unemployment and low levels of disposable income being cited in two [33, 35], short sleep duration being cited in three [38, 42–45], and lack of physical exercise being cited in five studies [15, 19, 40]. Thus, it is confirmed that these factors are those most associated with effects on mental health conditions in terms of cross-sectional and longitudinal direction.

In terms of SES factors such as disposable income and employment status, previous studies reported that lower income groups showed an association with poor mental health conditions [32–35], and that the socioeconomically disadvantaged were more likely to experience poor mental health compared with the advantaged [32]. In corroboration, individuals who had faced financial hardship showed significantly poorer mental health conditions [32, 36]. Nevertheless, the results from our study based on the longitudinal panel data showed that variations within the same variable, such as between higher and lower levels of disposable income, did not cause any changes in mental health conditions. When comparing the FE-CLR model with the RE-CLR model analysis among all participants, the middle level of disposable income was significantly associated with a poor mental health condition exclusively in the RE-CLR model, which was not supported by the Hausman test. This implies that the estimated associations in the RE-CLR model were mainly caused by participants' time-invariant factors. Those time-invariant factors were all controlled in the FE-CLR model where no significant association was found between disposable income and mental health condition. Similarly, among men, the low level of disposable income was significantly associated with a poor mental health condition exclusively in the RE-CLR model, which was not supported by the Hausman test. From these results, we can interpret that poor mental health conditions tend to be observed among men and women with middle levels of disposable income and among men with low levels of disposable income. While lower levels of disposable income may not be the cause of poor mental health conditions, it is a potential risk factor that can result in poor mental health.

Regarding employment status, the results of our study showed that both unemployed all samples and male samples had significant associations with poor mental health conditions. Lindstrom et al. presented that the unemployed and participants during periods of long-term sick leave had significantly higher ratios of poor psychological well-being among both men and women by using multiple analyses in a cross-sectional study [32]. A likely supposition is that the unemployed and those experiencing long-term sick leave may have low levels of disposable income. In our study, SES factors including unemployment and low levels of disposable income were associated with poor mental health. The results implied that clarifying the link between the complex socioeconomic factors such as disposable income that may lead to poor mental health is a clear requirement to improve mental health conditions and to minimise health inequalities.

However, regarding employment status, the females' result was different from the males'. This study revealed that women who were self-employed or non-regular employees showed inverse associations with poor mental health conditions. In that regard, previous studies presented that non-permanent employees reported higher job dissatisfaction [33], but lower levels of stress than permanent employees [33, 34]. Further research is needed to identify the relationship between mental health and employment status by gender, particularly related to types of employment, gender roles, and their possible associations.

In terms of lifestyle factors, sleep duration on weekdays and physical exercise habits for men and women alike were significantly associated with poor mental health in both the RE-CLR and FE-CLR models. Specifically, sleeping less than 6 hours a night significantly contributed to poor mental health outcomes compared with sleeping more than 6 hours a night. In correlation, previous studies showed significant negative impacts of short sleep duration among generations currently in their 'working years' on depression [35, 36] and anxiety [37, 38]; poorer GHQ-12 scores were demonstrated as well [38]. Thus, short sleep duration could be a cause of poor mental health [37], and our findings suggest that sleeping 6 hours or more per day can be critical for improved mental health outcomes.

Regarding exercise, the lack of physical activity could be a risk factor for poor mental health conditions [39], and some studies report that physical activity could play a significant role in promoting better mental health outcomes [15, 19, 39–41]. Moreover, one previous study reported that adolescents who had low aerobic fitness were more likely to report poorer sleep quality [42]. Based on these findings, there is a need to promote regular practice of physical activities in order to improve quality of sleep and mental health conditions, mainly because it helps reduce stress levels.

Among men, living alone showed a significant association with poor mental health exclusively in the RE-CLR model. We can interpret these results as poor mental health conditions are likely to be observed in men who live alone, but the results do not allow for an understanding that a change from living with someone to living alone may influence mental health, because the FE-CLR model did not show a significant relationship between living alone and mental health. Some previous cross-sectional studies presented that depressive symptoms were significantly associated with living alone [35, 43, 44]. Still, according to our results based on longitudinal panel data, there was no significant longitudinal and possibly causal association between living alone and poor mental health.

Regarding drinking habits among all participants, the RE-CLR model indicated that drinking alcohol more than 3 times per week had significant inverse associations with poor mental health conditions in the RE-CLR model, but it did not show any significant association in the FE-CLR model, which was supported by the Hausman test. These results imply the reserve causality. That is, drinking habits (drinking alcohol more than 3 times per week) do not necessarily improve mental health, but participants with a good mental health condition tended to

drink alcohol more than 3 times per week. The association between mental health and drinking habits are controversial. A study reported that problematic drinking habits are associated with depressive symptoms and suicidal ideation for men and women alike [35]. Notwithstanding, some studies, including a cohort study, reported that there was no association between mental health and drinking habits [15, 19, 39]. For this study, we controlled for unobservable and time-invariant variables, such as natural good mental health conditions, by using the FE-CLR models. Then, we found that there is no longitudinal association between mental health and drinking habits, a finding consistent with previous cohort studies [15, 19, 39]. Contrastingly, problem drinking may be a part of some stress coping strategies [45], so future studies are warranted to identify the relationship between stress coping and the amount of alcohol consumption.

Past research showed an association between practising several healthy lifestyle activities daily and reducing the risk of depression [13]. This same study suggested that sleeping for a proper duration, engaging in regular physical activity, and following abstemious drinking habits, which are all healthy lifestyle practices, could be essential for improving mental health. However, the underlying mechanisms that relate SES and lifestyle-related factors with mental health are complex [46], and studies that enhance our understanding related to these mechanisms are warranted, mainly studies that allow for finding causal relationships between variables, such as randomised control trials.

This study has several limitations. First, since we utilised a panel dataset, there is a possibility of sample attrition bias. In that regard, Baltagi reported that the average attrition rate of panel studies revolves around 10% [25]. In corroboration, our study showed an attrition rate revolving less than 10%. Further, in our analysis, we need to consider the possibility that respondents who dropped out might have had unhealthy outcomes or have been in unfavourable situations [25]. Future studies that provide other types of longitudinal data and that ensure consistency in terms of participants are warranted. Second, the questionnaires about SES and lifestyle factors were self-reported. Although we analysed data using the RE-CLR and FE-CLR models, under- or over-reporting of socially desirable attitudes and recall bias are a reality for this type of measure, and future studies should address these problems. Third, regarding causal relationships, we only controlled for the time-invariant factors using fixed-effects models. Thus, if any time-variant factors brought about reverse causality, our estimated results do not allow for the understanding of a causal relationship between two variables. Future studies should address this issue and control for time-variant factors that can bring reverse causality.

However, this study successfully examined associations between mental health outcomes and their potential risk factors using a large-scale, 5-wave longitudinal panel dataset which has some benefits. First, panel data suggests that individuals are heterogeneous and enables control of unobservable and time-invariant variables. For that, the RE-CLR and FE-CLR models were employed to identify longitudinal associations and the results of the Hausman test identified which model was a better fit for analysing socioeconomic and lifestyle-related factors' effects on mental health outcomes. Contrastingly, cross-sectional studies cannot control for heterogeneity and suffer from omitted variable biases [25]. Second, panel data are better to analyse changes and to identify associations among respondents. Supported by previous studies, our analyses utilised panel data to examine how respondents' changes in SES- and lifestyle-related factors affected mental health outcomes [47]. Third, our respondents were taken from an age category of working adults, randomly sampled from a nationwide population. Our final number of observations exceeded 14,717, and participants' response rates were over 90% for every year. After controlling for relevant covariates, this study revealed probable risk factors for poor mental health conditions and the significant associations between socioeconomic and lifestyle factors and mental health.

## Conclusions

This study revealed that the prevalence of poor mental health conditions among women was higher compared to men during 2014–2018, the five years on which we focused. When comparing the RE-CLR and the FE-CLR models, poor mental health conditions tended to be observed among all respondents with middle levels of disposable income and among men with a low level of disposable income or who lived alone. Our results imply that there is no causal relationship between the levels of disposable income and mental health conditions, and that changes in disposable income between high and low levels would not cause any changes in mental health conditions. In this case, aggressive policy interventions for increasing disposal income for people with lower socioeconomic status would not be necessarily effective to minimize health inequalities. Regarding living status among men, we found no significant causal association between living alone and poor mental health conditions, so living alone was not shown to be a cause of poor mental health conditions. The results of this study showed that while lower levels of disposable income and living status are not significant causes, they are potential risk factors for poor mental health. Further research is needed to identify the effects of disposable income and living status on mental health conditions, as well as the role of gender.

SES factors such as unemployment showed significant associations with poor mental health conditions among men. However, among women, self-employed and non-regular employees showed inverse associations with poor mental health conditions. Future studies should focus on identifying the associations between mental health and employment status by gender, particularly related to the types of employment, cultural gender roles, and their possible associations.

Our results suggest that unhealthy lifestyle factors such as short sleeping duration and lack of physical exercise may be potential risk factors for poor mental health for both men and women. Also, we found no causal association between alcohol habits and mental health conditions. Based on our results and the results of previous studies, promoting healthy lifestyle practices would help improve mental health conditions. Thus, we recommend conducting more longitudinal studies to examine causal relationships.

There is a complex mechanism behind the association between socioeconomic and health-related factors that lead to health inequality. Further research should explore differences provoked by gender, age, SES, and healthy lifestyle practices on health outcomes. Determining the complex mechanisms that relate mental health conditions to socioeconomic and lifestyle factors can be beneficial, as this knowledge may allow us to develop effective social welfare policies and health promotion interventions that are adequately equipped to improve mental health conditions, thereby minimising health inequalities caused by socioeconomic factors.

## Acknowledgments

The authors gratefully acknowledge the time and effort given by the participants of this study.

## Author Contributions

**Conceptualization:** Isamu Yamamoto.

**Formal analysis:** Miwako Nagasu.

**Funding acquisition:** Isamu Yamamoto.

**Investigation:** Miwako Nagasu.

**Methodology:** Miwako Nagasu.

**Project administration:** Isamu Yamamoto.

**Supervision:** Isamu Yamamoto.

**Writing – original draft:** Miwako Nagasu, Isamu Yamamoto.

**Writing – review & editing:** Miwako Nagasu, Isamu Yamamoto.

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
