## [Decision Letter · Decision Letter 0]

15 May 2020

PONE-D-20-10688

Impact of socioeconomic- and lifestyle-related risk factors on poor mental health conditions: A nationwide longitudinal 5-wave panel study in Japan

PLOS ONE

Dear Dr. Nagasu,

Thank you for submitting your manuscript to PLOS ONE. After careful consideration, we feel that it has merit but does not fully meet PLOS ONE’s publication criteria as it currently stands. Therefore, we invite you to submit a revised version of the manuscript that addresses the points raised during the review process.

The three reviewers addressed several major and minor concerns about your manuscript. Please revise your manuscript carefully.

We would appreciate receiving your revised manuscript by Jun 29 2020 11:59PM. To enhance the reproducibility of your results, we recommend that if applicable you deposit your laboratory protocols in protocols.io, where a protocol can be assigned its own identifier (DOI) such that it can be cited independently in the future. For instructions see: http://journals.plos.org/plosone/s/submission-guidelines#loc-laboratory-protocols

We look forward to receiving your revised manuscript.

Kind regards,

Kenji Hashimoto, PhD

Academic Editor

PLOS ONE

2.PLOS requires an ORCID iD for the corresponding author in Editorial Manager on papers submitted after December 6th, 2016. Please ensure that you have an ORCID iD and that it is validated in Editorial Manager. To do this, go to ‘Update my Information’ (in the upper left-hand corner of the main menu), and click on the Fetch/Validate link next to the ORCID field. This will take you to the ORCID site and allow you to create a new iD or authenticate a pre-existing iD in Editorial Manager. Please see the following video for instructions on linking an ORCID iD to your Editorial Manager account: https://www.youtube.com/watch?v=_xcclfuvtxQ

Reviewers' comments:

Reviewer's Responses to Questions

**Comments to the Author**

1. Is the manuscript technically sound, and do the data support the conclusions?

Reviewer #1: No

Reviewer #2: Yes

Reviewer #3: Yes

2. Has the statistical analysis been performed appropriately and rigorously? 

Reviewer #1: I Don't Know

Reviewer #2: Yes

Reviewer #3: Yes

3. Have the authors made all data underlying the findings in their manuscript fully available?

Reviewer #1: Yes

Reviewer #2: Yes

Reviewer #3: Yes

4. Is the manuscript presented in an intelligible fashion and written in standard English?

Reviewer #1: Yes

Reviewer #2: Yes

Reviewer #3: Yes

5. Review Comments to the Author

Reviewer #1: Dear authors and editors

Thank you for the opportunity to review this study. The association between socioeconomic position and lifestyle and the impact they have on mental health are important issues in order to direct actions towards improving mental health. The authors use repeated answers to the GHQ from national representative surveys over a five-year period. There are two aims; one to examine the prevalence of poor mental health and secondly to analyse the differences between the estimates from random and fixed models of logistic regression, in order to identify risk factors for poor mental health.

The abstract teases with the statement, that the results imply the reverse causality that poor mental health conditions cause lower disposal income. Usually it is considered a fact, that serious mental disorders (schizophrenia/ bipolar disorder) causes low socioeconomic position whereas common mental disorders are caused by low socioeconomic position, therefor the study draw my attention.

Unfortunately, I find the manuscript hard to evaluate, due to many discrepancies in tables and the text as well as methodological problems. I cannot recommend publication. In the following you have my comments to the major problems in the study. I hope the comments are useful for you.

Abstract

The abstract gives a good introduction but should state the aims clearly and mirror them in the reported results – e.g. prevalence is not mentioned at all.

Introduction

The authors quote WHO for stating inequality in health is unfair and subsequent studies have found …. The studies referred to are much before the statement by WHO (ref 4 & 5).

Several longitudinal studies in this field are not included in the literature mentioned. It is claimed the cross-sectional studies are unadjusted – that is quite rare, at least for studies on common mental disorders.

The negative impact on mental health of economic changes are well documented – one cited here. 1

Methods

JPHS is short of Japanese Household Panel Survey, I guess KHPS is short of Keio Household Panel Survey?

It is not evident how you reach 3.501 participants out of – as I can calculate (4.005 + 10.458) – 2.204. Then you have a drop out of 8.758? Even though you have a response rate of 91-98% in 2014 – 2018 in table 1 – which I do not understand at all. Here you have much more respondents: 14.717? And we have a division of JHPS and KHPS – even though they have merged/renamed in 2009? We need a clearer presentation of how the sample ends at 3.501 and a revision of Table 1, where n respondents is replaced by received questionnaires, if that is so – “waves” need to be explained as well and why both KHPS and JHPS appears. If none are replaced in the five years, how many drop out?

The variables must have been defined by JHPS in the 2004, so not much to do about the questions and categorisation, but it would be informative to know if the income-groups are equally shared in Japan with 1/3 in each? Why these income-groups?

Disposable household income is a good measure, if the size of the household is known; here we only have one or more than one. In other words: we know the vehicle has 100 HP, but we do not know the load it has to carry - if it is a lorry or a car.

The method of measuring health behaviors seems fair from a health perspective, except for the question on drinking. This question is not comparable to other studies on drinking habits/alcohol use and does not seem relevant. To drink a beer three times a week is not considered a drinking problem in most OECD countries. It may be different in Japan, but then it needs explanation.

The chosen statistical methods are unconventional.

Results

Here 14.717 participants are presented as shown in table two. What to believe?

Table 2: %SD – do you mean %? Very few women have regular employment and drink alcohol three times a week, very few respondents have a low household income. It needs an explanation – the numbers after “in Japanese Yen” what are they? The “0 points – 12 points” at GHQ, what is that? An average? The GHQ- score is a result pooled over the years I guess, please write this, if so. 42% women have poor mental health? It makes you wonder if GHQ valid in this context? Please add reference on the validation of GHQ-12 in a Japanese context in the method section.

It is not mentioned in the method section, but the adjustments are done only for other variables than the one presented/analysed, isn’t it?

Table 3: The table is difficult to read. I would suggest the FE/RE was stated in the heading section. Bold is stated to indicate statistically significant findings but seems used at random. How can an AOR for RE-female alcohol >= 3 times a week at 0.752 (0.547 – 1.179) be significant? The non-significant results for men and women end up being 0.7885 (0.629 – 0.979)? Why are there fewer observations for the FE-anaysis than the RE-analysis?

Discussion

Limitations. Dropouts are discussed in general terms. What was the actual characteristics of the dropouts in this study? The income variable is very screwed – is that sampling bias or at true reflection of the study population?

Information bias is mentioned only as recall. However, the instrument is not (or is?) validated; again, when adjusting for the number of persons in household – which is very good -the variable is not covering that, but only one or more than one. This is a serious problem and makes any conclusion related to income invalid.

“Further research is needed to identify the effects of disposable income and living status on mental health conditions, as well as the role of gender”. They do exist in plenty – as do so for employment status and gender.

You do not reflect on the high prevalence of poor mental health, why? it is a central study objective. However, to examine the prevalence of poor mental health longitudinal data are not optimal, unless you want to give information on the development in mental health. Again, it is not evident if GHQ is validated in a Japanese population – and a pooled prevalence of 42% women in poor mental health does not seem reliable.

As for household income as socioeconomic index the validity is poor – first very few are in lowest category and even though it is stated the analyses are adjusted for number of persons in the household, when in fact it is only adjusting for one or more than one.

As for the FE logistic regression analyses vs RE logistic analyses none of them account for the time each individual contributes with in the time series (person-years), and thus less accurate than the traditional methods used for longitudinal studies.

1. Barbaglia MG, M. tH, Dorsselaer S, et al. Negative socioeconomic changes and mental disorders: a longitudinal study. J Epidemiol Community Health 2015;69(1):55-62.

Reviewer #2: In general, this study is well constructed to gain a rational outcome. I have some minor comments to be addressed for better understanding of the study.

In the introduction section, the author mentioned that Japan was shown to be one of the countries with the highest suicide rates in the world. It is partially acceptable, but a little over-represented. Some Eastern European nations as well as Russia has higher suicide rate. Also, in latest statistics, those of the US and Sweden are not so different from Japan's. I am afraid that outdated articles the author referred can be misleading.

In this study, the author classified some items the participants answered into some groups. How the author decide the thresholds of each group?

For example, are there any reasons that people taking 6-7 hours of sleep in a day is different to those taking >7 hours sleep, not >8? In my understanding, there are rich evidence suggesting sleeping under 6 hours in a day is harmful. But how many hours you should sleep is controversial.

Also, the author referred a Glozier's work. But its subjects were limited to young people recommended to take 8-9 hours sleep. The author should consider to show better preceding studies.

In this study, only 7.9% of the participants lived alone. According to recent official statistics in Japan, one-fourth of the household is composed of one person. In my calculation, single person household was 10.7% in 2014. Did the author compare the demographic data with those of contemporary official statistics? If there is a large discrepancy between them, the representativeness of the panel is doubtful.

The same thing is also adapted to employment status, but it seems consistent with the official statistics, as far as my checking.

Table 2 is difficult to read at a glance because each number and the mean share the same column. The author should rearrange it.

Is the mean income of Japanese only 542.8 JPY?

What does "0 point - 12 points" mean in the GHQ score section?

Figure 1. is hardly understandable and space-killing. What does the vertical axis mean? I reckon it shows the percentage. Anyway, it seems better to choose another type of graph (polygonal line graph, or a chart maybe).

Reviewer #3: This epidemiological survey is important for mental healthy field, worth reading. The authors however shoud propose the hypothesis of the study clearly, and enphasize the new findings in the present study, and the differences from the previous reports throuout the manuscript.

6. PLOS authors have the option to publish the peer review history of their article (what does this mean?). If published, this will include your full peer review and any attached files.

Reviewer #1: Yes: Aake Packness, PhD, MPH, RN

Reviewer #2: Yes: Akihiro Shiina

Reviewer #3: Yes: Yoshimura Reiji

---

## [Author Response · Author response to Decision Letter 0]

10 Sep 2020

Authors’ response to reviewers:

PLOS ONE

PONE-D-20-10688

Impact of socioeconomic- and lifestyle-related risk factors on poor mental health conditions: A nationwide longitudinal 5-wave panel study in Japan

Reviewer #1: 

Thank you very much for reviewing our manuscript. We have revised the manuscript according to your comments and suggestions. 

Abstract

1. The abstract gives a good introduction but should state the aims clearly and mirror them in the reported results – e.g. prevalence is not mentioned at all.

Response: 

We have rewritten many parts of the abstract and added the following sentences. Further, the results of this study are added in the abstract. 

‘This study aims to reveal the prevalence of poor mental health conditions among Japanese individuals and to identify SES- and lifestyle-related risk factors that might lead to these conditions.’ 

‘The prevalence of poor mental health conditions, represented by a GHQ-12 score of 4 or more, was 36.1% and 42.0% of men and women, respectively.’

‘Various factors, such as unemployment, low household income, short nightly sleeping duration, and lack of exercise, showed significant longitudinal (within) associations with mental health conditions estimated by the FE-CLR models.’ 

Introduction

2. The authors quote WHO for stating inequality in health is unfair and subsequent studies have found …. The studies referred to are much before the statement by WHO (ref 4 & 5).

Response: 

Thank you for pointing this out. We have corrected the sentence as follows: 

‘Several studies have found that differences in socioeconomic status (SES) are one of the causes of health inequalities and are associated with health outcomes (2, 4, 5).’

We wanted to cite these references because, during the 20 years, surprisingly, we have not solved health inequality and still WHO has to write ‘Health inequality is unfair’. As you mentioned, a study (Reference No. 5, 2003) reported that suicide risk is strongly associated with factors related to low SES, such as unemployment and low income. Lynch et al (Reference No. 4, 1997) also mentioned that socioeconomic inequalities in health have often made reference to the observation that poor health behaviours and psychosocial characteristics cluster in low SES groups. Since then, even in 2017, we should realise that still WHO had to write ‘Health inequalities are unfair’ on their website. We believe that these references should be included to discuss this issue. 

3. Several longitudinal studies in this field are not included in the literature mentioned. It is claimed the cross-sectional studies are unadjusted – that is quite rare, at least for studies on common mental disorders.

Response: 

Thank you for pointing this out. We added some references of longitudinal studies as follows: 

‘Thus, any possible associations between suicide and mental health, SES, and lifestyle-related factors should be investigated (11, 12).’

‘Cross-sectional and longitudinal studies have demonstrated that healthy lifestyle factors are positively associated with better mental health outcomes (13). Thus, the relationship between mental health conditions and lifestyle factors, such as sleeping duration (14, 15), habitual physical exercise (16, 17), smoking habits (18, 19), and alcohol consumption (20), have been investigated to identify methods for preventing mental illness.’

As the mentioned by the reviewer that it is claimed the cross-sectional studies are unadjusted – that is quite rare, at least for studies on common mental disorders. We partially agree with the comment. We believe that this does not imply ‘the cross-sectional studies’, about ‘the logistic regression model used in the cross-sectional studies’, but this implies that the variables are adjusted in the logistic regression models and so on. 

As we explained, we applied fixed-effects regression models that automatically adjust for all time-constant unobserved confounders and help reduce the risk of omitted variable bias, as well as adjust for identified time-varying confounders. These characteristics are different from those of logistic regression models. 

Accordingly, we added the following sentence as an explanation about the fixed-effects model:

‘The fixed-effects regression models are used to adjust for all time-constant unobserved confounders and decrease the risk of omitted variable bias, as well as adjust for identified time-varying confounders.’

4. The negative impact on mental health of economic changes are well documented – one cited here. 

Response: 

Thank you for the suggestion. We added the recommended paper as follows: 

‘Barbaglia et al. reported that negative socioeconomic changes, such as substantial reduction in household income and job loss, significantly increased the risk of incident mental disorders (23).’

Methods

5. JPHS is short of Japanese Household Panel Survey, I guess KHPS is short of Keio Household Panel Survey? It is not evident how you reach 3.501 participants out of – as I can calculate (4.005 + 10.458) – 2.204. Then you have a drop out of 8.758? Even though you have a response rate of 91-98% in 2014 – 2018 in table 1 – which I do not understand at all. Here you have much more respondents: 14.717? And we have a division of JHPS and KHPS – even though they have merged/renamed in 2009? We need a clearer presentation of how the sample ends at 3.501 and a revision of Table 1, where n respondents is replaced by received questionnaires, if that is so – ‘waves’ need to be explained as well and why both KHPS and JHPS appears. If none are replaced in the five years, how many drop out?

Response: 

As we mentioned in abbreviation list, the JHPS/KHPS stands for the Japan Household Panel Survey and Keio Household Panel Survey. The JHPS/KHPS has two series of samples: the KHPS and the JHPS started in 2004 and in 2009, respectively. In 2014, the KHPS and JHPS were integrated in one survey called the KHPS/JHPS. Given that the JHPS and KHPS comprise the same questionnaire from 2014, we can use both samples for the analysis. Hence, we added the sentence below mentioned in the method section:

‘The JHPS/KHPS is a panel (longitudinal) data in which there are multi-dimensional data involving measurements over time. It contains time-series observations of each participant, and multiple phenomena (answers for questionnaires in this case) are obtained over multiple time periods (for 5 years in this study) for the same participants’.

We also wrote the following in the abstract: ‘Our sample comprised 14,717 observations of 3,501 individuals aged 22–59 years for five waves of the Japanese Household Panel Survey.’. Regarding the number of participants who dropped out from the survey, we added the numbers in brackets in Table 1. 

We discussed the number of respondents who participated in the studies from 2014 to 2018 in Table 1. Regarding your calculation, we apologize, but we cannot understand how you derived these numbers in Table 1: (4.005 + 10.458) – 2.204. In the table, we also added information on how to calculate the response rate (%). 

When using panel data or longitudinal data in Public Health 1,2) and Economics, researchers commonly use technically the word ‘wave’ instead of ‘year’. However, we added an explanation about ‘wave’ in sampling of the respondents section as follows: ‘In panel survey, ‘wave’ is generally used as the same meaning as ‘year’

Reference: 

1: Carroll SJ, Dale MJ, Niyonsenga T, Taylor AW, Daniel M. Associations between area socioeconomic status, individual mental health, physical activity, diet and change in cardiometabolic risk amongst a cohort of Australian adults: A longitudinal path analysis. PloS one. 2020;15(5):e0233793.

2: Wang S, Mak HW, Fancourt D. Arts, mental distress, mental health functioning & life satisfaction: fixed-effects analyses of a nationally-representative panel study. BMC public health. 2020;20(1):208.

Table 1. Number of respondents and response rates for each survey and number of recruited participants from each survey.

JHPS KHPS Total 

Wave Survey year Total number of respondents1) Response rate (%)2) Respondents in this study Wave Survey year Total number of respondents Response rate (%) Respondents in this study 

6 2014 2,358 (238) 91.1 1,385 11 2014 3,312 (275) 92.6 2,102 3,487

7 2015 2,198 (186) 93.0 1,253 12 2015 3,124 (229) 98.8 1,939 3,192

8 2016 2,048 (163) 92.8 1,144 13 2016 2,945 (207) 94.0 1,776 2,920

9 2017 1,885 (175) 91.9 1,057 14 2017 2,741 (229) 92.7 1,628 2,685

10 2018 1,742 (147) 92.2 970 15 2018 2,549 (206) 93.0 1,463 2,433

Total 10,231 (909) 92.2 5,809 Total 14,671 (1,146) 94.2 8,908 14,717

1) Total number of respondents: the number of collected questionnaires that were completed by the respondents. The numbers in brackets show the number of people who dropped out. 

2) Response rate (%) = (number of completed questionnaires – number of people who restarted answering questionnaire)/number of completed questionnaires in the previous year) × 100)

6. The variables must have been defined by JHPS in the 2004, so not much to do about the questions and categorisation, but (1) it would be informative to know if the income-groups are equally shared in Japan with 1/3 in each? (2) Why these income-groups?

Response:

Thank you for your question. The categorisation is the same as the Japanese government survey, the National Survey Health and Nutrition (Ministry of Health, Labour and Welfare, Japan) in 2014. The income groups are not equally shared with 1/3 in both the National Survey Health and Nutrition and JHPS, as presented in Table 2. It should be noted that the table shows that the proportion of each income group is different between the government survey and JHPS. The partial possible reason for this difference is that JHPS sample include more single person household because the participants of JHPS were selected at the individual level while those of the government survey were at the household level. 

Table 2. Categories of disposable income

 National Survey Health and Nutrition (2014) This study (JHPS)

 Households All 

 n % n %

≥ 6,000 K 717 22.0 4856 36.9

2,000 K-< 6,000 K 1,765 54.0 7373 56.0

< 2,000 K 784 24.0 935 7.1

Total 3,266 100.0 13164 100.0

To account for the possible bias due to the smaller sample of single persons, we conducted a weighted logit regression using the inverse of the number of households as weight. As presented in Table 3, although some of the statistical significance vary, the sign (more than 1 or not) and magnitude of the odds ratio seem not to be so different between the estimates with and without weight. Thus, we regard that although there is a sampling bias in our data because less persons living alone were selected, it does not have large effects on the estimation results.

Table 3. Results of RE and estimation with weight

　 　 Results of RE　 Results of estimation with weight 

　 　 odds ratios odds ratios

Sex Men Refe Refe 

 Women 1.419*** 1.236***

 (1.107–1.819) (1.122–1.362)

Age Under 39 Refe Refe 

 40–49 0.971 0.951

 (0.791–1.190) (0.866–1.045)

 50–59 0.836 0.861***

 (0.666–1.049) (0.782–0.948)

Number of persons in the household ≥ 2 people Refe Refe 

 One person 1.286 1.248***

 (0.946–1.749) (1.091–1.428)

Employment status unemployed 1.501*** 1.165**

 (1.122–2.008) (1.022–1.328)

 self-employed 0.832 0.832***

 (0.626–1.105) (0.737–0.940)

 regular employee Refe Refe 

 non-regular employee 0.983 0.987

 (0.774–1.247) (0.887–1.098)

Disposable income per household ≥ 6,000 K Refe Refe 

 2,000 K-< 6,000 K 1.254*** 1.198***

 (1.075–1.464) (1.106–1.299)

 < 2,000 K 1.977*** 1.770***

 (1.475–2.650) (1.517–2.064)

Sleep duration of the respondent ≥ 7 h Refe Refe 

 6–7 h 1.100 1.028

 (0.951–1.272) (0.945–1.119)

 < 6 h 1.644*** 1.418***

 (1.355–1.993) (1.288–1.561)

Physical exercise ≥ 3 days/week Refe Refe 

 ≤ 2 days/week 1.175 1.228***

 (0.897–1.538) (1.054–1.429)

 No exercise 1.637*** 1.449***

 (1.284–2.088) (1.270–1.652)

Smoking habit Never Refe Refe 

 Quit 1.135 1.076

 (0.909–1.419) (0.979–1.183)

 Sometimes + everyday 1.217 1.126**

 (0.950–1.560) (1.020–1.243)

Drinking alcohol Never Refe Refe 

 ≤ 2 times/week 0.893 0.879***

 (0.745–1.069) (0.806–0.959)

 ≥ 3 times/week 0.785** 0.894**

 (0.629–0.979) (0.812–0.985)

 Constant 0.189*** 0.360***

 (0.129–0.278) (0.301–0.431)

 Observations 12,681 12,681

　 Number of respondents 3,359 　

Robust cieform in parentheses 

*** p<0.01, ** p<0.05, * p<0.1 

To explain the above points, we added the following sentences into the method and limitation part: 

‘The categorisation method is the same as the government survey, National Survey Health and Nutrition (Ministry of Health, Labour and Welfare, Japan) in 2014.’

‘First, for the proportion of single person household, we compared the demographic data of this study with those of the national census taken on Oct. 2015. The result of this study seems to have considerably fewer participants lived alone than the results of the national census. Because the participants of this study were selected at the individual level, that of the national census were at the household level. We also calculated the composition of single-person living at the individual level as 14.4% (the number of single person household is 18,420, which amounts to 14.4% of the Japanese population of 128,000 thousands). Therefore, there is about a double difference between population (14.4%) in the national census and our sample (7.9%). To account for the possible bias due to the smaller sample of single-person, we conducted a weighted logit regression using the inverse of the number of households as weight. Although there are some statistical significance between RE-CLR and RE-CLR with weight, the sign (more than 1 or not) and magnitude of the odds ratios seem not to be so different between the estimates with and without weight. Thus, we regard that although there is a sampling bias in our data in the sense that less persons living alone were selected, it does not largely affect the estimation results.’

7. Disposable household income is a good measure, if the size of the household is known; here we only have one or more than one. In other words: we know the vehicle has 100 HP, but we do not know the load it has to carry - if it is a lorry or a car.

Response:

Thank you for your comment. Our intention to categorise into two groups was to distinguish between married or unmarried. As the number of household member is actually available in JHPS, we confirmed that the estimation results did not change significantly even if we used the dummy variables indicating the size of household: a one-person, two-person, three-person, and four-person dummies or more. Please find the comparison of the estimation results depending on two types of the variable about ‘the number of household members.’ 

To explain this, we added the following sentence in the Methods section.

‘Although the number of household members is actually available in JHPS, it was categorised into two groups: living with someone (≥ 2 people) and living alone. We confirmed that the estimation results did not change significantly even if we used the dummy variables indicating the size of household: a one-person, two-person, three-person, and four or more person dummies.’

Table 4. Estimated associations between the GHQ scores and risk factors with two types of the variable ‘the number of household member. 

　 　 　 　 　 　 　

 All samples All samples 

　 　 AOR(C.I.) 4) AOR(C.I.)4) 　 AOR(C.I.) 5) AOR(C.I.)5)

Sex Men Refe Refe 

 Women 1.419*** 1.419*** 

 (1.107 - 1.819) (1.108 - 1.818) 

Age Under 39 Refe Refe Refe Refe 

 40-49 0.971 1.117 0.968 1.093

 (0.791 - 1.190) (0.815 - 1.532) (0.790 - 1.186) (0.800 - 1.495)

 50-59 0.836 1.082 0.819 1.067

 (0.666 - 1.049) (0.693 - 1.691) (0.651 - 1.029) (0.685 - 1.661)

Number of persons in the household 1 person 1.286 0.999 1.396 0.860

 (0.946 - 1.749) (0.634 - 1.575) (0.962 - 2.026) (0.495 - 1.492)

 2 people Refe Refe 1.184 0.854

 (0.873 - 1.608) (0.529 - 1.378)

 3 people 1.167 0.926

 (0.884 - 1.542) (0.618 - 1.387)

 4 people 1.052 0.924

 (0.813 - 1.361) (0.643 - 1.328)

 5 people Refe Refe

 6 people 1.038 1.076

 (0.676 - 1.594) (0.595 - 1.945)

 ≥ 7 people 0.852 0.940

 (0.486 - 1.496) (0.406 - 2.173)

Employment status Unemployed 1.501*** 1.727** 1.491*** 1.692**

 (1.122 - 2.008) (1.134 - 2.629) (1.115 - 1.994) (1.112 - 2.574)

 Self-employed 0.832 1.193 0.830 1.163

 (0.626 - 1.105) (0.727 - 1.958) (0.625 - 1.102) (0.712 - 1.898)

 Regular employee Refe Refe Refe Refe 

 Non-regular employee 0.983 0.958 0.985 0.950

 (0.774 - 1.247) (0.680 - 1.350) (0.776 - 1.250) (0.675 - 1.337)

Disposable income per household ≥ 6,000 K Refe Refe Refe Refe 

 2,000 K-< 6,000 K 1.254*** 1.118 1.243*** 1.109

 (1.075 - 1.464) (0.924 - 1.352) (1.065 - 1.451) (0.918 - 1.339)

 < 2,000 K 1.977*** 1.441* 1.940*** 1.423*

 (1.475 - 2.650) (0.993 - 2.092) (1.446 - 2.602) (0.981 - 2.065)

Sleep duration of the respondent ≥ 7 h Refe Refe Refe Refe 

 6-7 h 1.100 1.128 1.094 1.116

 (0.951 - 1.272) (0.952 - 1.335) (0.947 - 1.265) (0.943 - 1.321)

 < 6 h 1.644*** 1.503*** 1.624*** 1.466***

 (1.355 - 1.993) (1.177 - 1.919) (1.339 - 1.970) (1.148 - 1.871)

Physical exercise ≥ 3 days/week Refe Refe Refe Refe 

 ≤ 2 days/week 1.175 1.066 1.174 1.074

 (0.897 - 1.538) (0.792 - 1.435) (0.897 - 1.537) (0.797 - 1.447)

 No exercise 1.637*** 1.418** 1.640*** 1.432**

 (1.284 - 2.088) (1.062 - 1.895) (1.286 - 2.091) (1.072 - 1.912)

Smoking habit Never Refe Refe Refe Refe 

 Quit 1.135 0.929 1.130 0.927

 (0.909 - 1.419) (0.579 - 1.490) (0.905 - 1.412) (0.577 - 1.488)

 Sometimes + everyday 1.217 0.940 1.215 0.943

 (0.950 - 1.560) (0.520 - 1.700) (0.948 - 1.557) (0.521 - 1.708)

Drinking alcohol Never Refe Refe Refe Refe 

 ≤ 2 times/week 0.893 1.033 0.898 1.032

 (0.745 - 1.069) (0.809 - 1.320) (0.750 - 1.076) (0.807 - 1.319)

 ≥ 3 times/week 0.785** 0.775 0.792** 0.774

 (0.629 - 0.979) (0.552 - 1.088) (0.635 - 0.989) (0.550 - 1.088)

 Constant 0.189*** 0.176*** 

 (0.129 - 0.278) (0.114 - 0.271) 

 Observations 12,681 5,992 12,737 6,025

 Number of respondents 3,359 3,359 

　 　 0.0061 　 　 　 　

1) Level of the GHQ 12 score (0: ≤ 3 points, 1: ≥ 4 points)

2) Robust cieform in parentheses

3) *** p<0.01, ** p<0.05, * p<0.1

4) Adjusted odds ratios (AOR) with 95% CI (adjusted for sex, age, number of persons in the household (2 categories: one person or ≥　2 people), employment status, disposable income per household, sleep duration in weekdays, physical exercise, smoking habit and drinking alcohol).

5) Adjusted odds ratios with 95% CI (adjusted for sex, age, number of persons in the household (7 categories: 1 person, 2 people, 3 people, 4 people, 5 people, 6 people, and 7 people and more), employment status, sleep duration in weekdays, physical exercise, smoking habit and drinking alcohol).

6) Refe: Reference 1

8. The method of measuring health behaviours seems fair from a health perspective, except for the question on drinking. This question is not comparable to other studies on drinking habits/alcohol use and does not seem relevant. To drink a beer three times a week is not considered a drinking problem in most OECD countries. It may be different in Japan, but then it needs explanation.

Response:

There are various approaches used to determine drinking habits. It would be ideal to ask the amount of alcohol directly and obtain an answer for the amount of alcohol content in a beverage. However, almost people do not know the alcohol content in liquor. Further, as it is well known that the alcohol content in beer varies, nutritionists attempted to develop simple questions such as frequency of alcohol intake1). Asking for the frequency of alcohol intake is a common method in nutrition. Compared to other approaches, such as asking for the amount of alcohol in 24 hours, this approach has some benefits. Some studies on alcohol content and frequency of alcohol intake have already been conducted2,3). In particular, the National Health and Nutrition Survey conducted by the Japanese government used questions about the frequency of alcohol intake. Researchers have been attempting to validate the food frequency questionnaire (FFQ)3). Therefore, we believe that this question is comparable to other studies and relevant to measure alcoholism of a person by using self-administrated questionnaire in this study. 

References: 

1: Willett W. Food frequency methods. Nutr Epidemiol. Third edition. Oxford, New York: Oxford University Press; 2012.

2:Sadakane A, Gotoh T, Ishikawa S, Nakamura Y, Kayaba K, Jichi Medical School Cohort Study G. Amount and frequency of alcohol consumption and all-cause mortality in a Japanese population: the JMS Cohort Study. Journal of epidemiology. 2009;19(3):107-15.

3: Harmouche-Karaki M, Mahfouz M, Obeyd J, Salameh P, Mahfouz Y, Helou K. Development and validation of a quantitative food frequency questionnaire to assess dietary intake among Lebanese adults. Nutrition Journal. 2020;19(1):65.

9. The chosen statistical methods are unconventional.

Response:

Thank you for the comment. However, we believe that the fixed-effects or random-effects model based on panel data are becoming standard statistical methods. These methods have been commonly and frequently used in economics literature and are used in other fields recently, including public health. For example, many published scientific paper used the fixed-effects and random-effects models to determine the association among potential risk factors in public health, such as the following. 

References:

1. Wang S, Mak HW, Fancourt D. Arts, mental distress, mental health functioning & life satisfaction: fixed-effects analyses of a nationally-representative panel study. BMC public health. 2020;20(1):208.

2. Wang J, Xu J, An R. Effectiveness of backward walking training on balance performance: A systematic review and meta-analysis. Gait & Posture. 2019;68:466-75.

3. Oshio T, Inoue A, Tsutsumi A. Associations among job demands and resources, work engagement, and psychological distress: fixed-effects model analysis in Japan. Journal of occupational health. 2018;60(3):254-62.

4. Oshio T. The association between individual-level social capital and health: cross-sectional, prospective cohort and fixed-effects models. Journal of epidemiology and community health. 2016;70(1):25.

5. Jin Q, Shi G. Meta-Analysis of SNP-Environment Interaction with Heterogeneity. Hum Hered. 2019;84(3):117-26.

6.　　　Kuroda S, Yamamoto I. Why Do People Overwork at the Risk of Impairing Mental Health? Journal of Happiness Studies. 2019;20(5):1519-38.

7.　　　Ocean N, Howley P, Ensor J. Lettuce be happy: A longitudinal UK study on the relationship between fruit and vegetable consumption and well-being. Social science & medicine. 2019;222:335-45.

8. Oshio T. The association between individual-level social capital and health: cross-sectional, prospective cohort and fixed-effects models. Journal of epidemiology and community health. 2016;70(1):25.

Results

10. Here 14.717 participants are presented as shown in table two. What to believe?

Response:

Would you please check the number in total (bottom of right corner) in Table 1? In the Abstract and Methods section, we wrote that 14,717 implies 14,717 observations/participants/questionnaires in total for 5-year questionnaire survey from 2014 to 2018. In econometrics, this is called 14,717 observations. 

We added the sentence as follows: ‘In total of this study, we used data from 14,717 observations of 3,501 individuals aged 22 to 59.’ 

11. 1) Table 2: %SD – do you mean %? 2) Very few women have regular employment and drink alcohol three times a week, very few respondents have a low household income. It needs an explanation – the numbers after “in Japanese Yen” what are they? 3) The “0 points – 12 points” at GHQ, what is that? 4) An average? 5) The GHQ- score is a result pooled over the years I guess, please write this, if so. 42% women have poor mental health? It makes you wonder if GHQ valid in this context? Please add reference on the validation of GHQ-12 in a Japanese context in the method section.

Response:

1) Thank you for the comment. %SD implies % and SD. We changed the layout accordingly as follows:

Table 2. Participants’ demographic characteristics.

 Total Men Women

Variables Group n % N % n %

Wave 

 Men 7,215 49.0% 

 Women 7,502 51.0% 

Number of persons in the household 

 ≥ 2 people 13,487 92.1% 6,415 89.4% 7,072 94.7%

 1 person 1,155 7.9% 757 10.6% 398 5.3%

Employment status 

 Unemployed 1,885 12.9% 271 3.8% 1,614 21.7%

 Self-employed 1,818 12.4% 1,106 15.4% 712 9.6%

 Regular employee 7,126 48.7% 5,332 74.3% 1,794 24.1%

 Non-regular employee 3,796 26.0% 466 6.5% 3,330 44.7%

Disposable income per household 

 ≥ 6,000 K 4,856 36.9% 2,424 37.2% 2,432 36.6%

 2,000 K-< 6,000 K 7,373 56.0% 3,720 57.1% 3,653 54.9%

 < 2,000 K 935 7.1% 370 5.7% 565 8.5%

Sleep duration. weekdays 

 ≥ 7 h 5,450 37.6% 2,710 38.2% 2,740 37.0%

 6–7 h 5,595 38.6% 2,769 39.1% 2,826 38.2%

 < 6 h 3,448 23.8% 1,607 22.7% 1,841 24.9%

Physical exercise 

 ≥ 3 days/week 1,415 9.7% 729 10.2% 686 9.2%

 ≤ 2 days/week 2,637 18.0% 1,518 21.2% 1,119 15.0%

 No exercise 10,568 72.3% 4,914 68.6% 5,654 75.8%

Smoking habit 

 Never 7,888 53.7% 2,524 35.0% 5,364 71.6%

 Quit 3,423 23.3% 2,220 30.8% 1,203 16.1%

 Sometimes + everyday 3,379 23.0% 2,459 34.1% 920 12.3%

Drinking alcohol habit 

 Never 5,192 35.4% 1,746 24.3% 3,446 46.2%

 ≤ 2 times/week 5,130 35.0% 2,415 33.6% 2,715 36.4%

 ≥ 3 times/week 4,328 29.5% 3,027 42.1% 1,301 17.4%

GHQ score 

 ≥ 4 points (poor) 5,713 39.1% 2,581 36.1% 3,132 42.0%

 ≤ 3 points 8,891 60.9% 4,562 63.9% 4,329 58.0%

 Mean (SD) Mean (SD) Mean (SD)

Age (years old) 45.3 (8.7) 45.2 (8.7) 45.4 (8.7)

Disposable income per household 5,428K (3,113K) 5,422K (2,959K) 5,433K (3,256K)

GHQ score 3.4 (3.4) 3.2 (3.4) 3.6 (3.4)

2) It was a unit of Japanese currency, and we deleted it.

3) We deleted ‘The 0 points – 12 points’.

4) It was not an average. It was a range of the GHQ score from 0 points to 12 points.

5) We deleted Figure 1 and made Table 3 as follows:

Table 3. Differences of GHQ scores of 4 points or more by sex, per year and pooled data.

Years Females Males P value 1)

2014 41.9% 35.8% ***

2015 41.9% 35.7% ***

2016 41.8% 37.9% *

2017 43.6% 36.9% ***

2018 40.7% 34.3% **

Pooled data 42.0% 36.1% n.s.

Table 3 shows the differences of GHQ scores of 4 points or more by sex per year and pooled data. As the reviewer recommended, we added ‘pooled data’ in the sentence as follows: 

‘In terms of pooled data of participants’ mental health conditions, 36.1% of men and 42.0% of women were shown to have poor mental health conditions (≥ 4 GHQ-12 score).’

According to the results of this study, 42% of female subjects obtained 4 or more points of GHQ. This implies 42% of women have poor mental health conditions. The results during 5 years, around 40% of women showed 4 and more points consistently. We have not found any reference in which GHQ is not valid for Japanese people. For example, Hori et al. wrote as follows1):

The GHQ-12 is a widely used, self-administered questionnaire that was originally designed as a screening tool for mental illness. The GHQ is also used in primary health care screening in the general population survey2). The GHQ-12 score was first applied to adults and then validated for adolescents as well3). Every item on the GHQ-12 describes a symptom and has four possible responses: the two answers which indicate the absence of the symptom are given a score of 0, and the other two which indicate the presence of the symptom receive a score of 1. The overall score on the scale will fall into a range of 0–12, with higher scores indicating more psychological distress. Good mental health was defined as a GHQ-12 score <4 and poor mental health as a score ≥4 4).

We added the sentences as follows into the method section: 

‘Previous study reported that they assessed the factor structure of the GHQ-12 for the Japanese general adult population. Data came from a sample of 1808 Japanese aged 20 years or older who were randomly selected based on the 1995 census (897 men and 911 women). Cronbach’s α coefﬁcients of GHQ-12 were 0.83 for Japanese men and 0.85 for Japanese women (32).’

References: 

1. Hori D, Tsujiguchi H, Kambayashi Y, Hamagishi T, Kitaoka M, Mitoma J, et al. The associations between lifestyles and mental health using the General Health Questionnaire 12-items are different dependently on age and sex: a population-based cross-sectional study in Kanazawa, Japan. Environ health prev med. 2016;21(6):410-21.

2. Doi Y, Minowa M. Factor structure of the 12-item General Health Questionnaire in the Japanese general adult population. Psychiatry Clin Neurosci. 2003;57(4):379-83.

3. Baksheev GN, Robinson J, Cosgrave EM, Baker K, Yung AR. Validity of the 12-item General Health Questionnaire (GHQ-12) in detecting depressive and anxiety disorders among high school students. Psychiatry research. 2011;187(1-2):291-6.

4. Goldberg DP, Oldehinkel T, Ormel J. Why GHQ threshold varies from one place to another. Psychological medicine. 1998;28(4):915-21.

12. It is not mentioned in the method section, but the adjustments are done only for other variables than the one presented/analysed, isn’t it?

Response:

Yes, it is. We have tried some adjustments with some models, but we presented only the model. 

13. 1) Table 3: The table is difficult to read. I would suggest the FE/RE was stated in the heading section. 

2) Bold is stated to indicate statistically significant findings but seems used at random. 3) How can an AOR for RE-female alcohol >= 3 times a week at 0.752 (0.547 – 1.179) be significant? 4) The non-significant results for men and women end up being 0.7885 (0.629 – 0.979)? 5) Why are there fewer observations for the FE-analysis than the RE-analysis?

Response:

1) About the table design, we changed the layout based on your suggestions as follows: 

Table 4. Estimated associations between participants’ General Health Questionnaire 12-item scores and risk factors by gender based on the random-effects conditional logistic regression models, on the fixed-effects conditional logistic regression models and on the Hausman tests.

 All samples Men Women 

 AOR(C.I.) 1) AOR(C.I.)2) AOR(C.I.)2) AOR(C.I.)2) AOR(C.I.)2) AOR(C.I.)2)

Model Type RE FE* RE FE* RE* FE

Sex Men Ref. 

 Women 1.419*** 

 (1.107 - 1.819) 

Age Under 39 Ref. Ref. Ref. Ref. Ref. Ref. 

 40-49 0.971 1.117 0.857 1.121 1.146 1.120

 (0.791 - 1.190) (0.815 - 1.532) (0.623 - 1.179) (0.692 - 1.815) (0.880 - 1.492) (0.738 - 1.700)

 50-59 0.836 1.082 0.766 0.935 0.936 1.184

 (0.666 - 1.049) (0.693 - 1.691) (0.538 - 1.092) (0.474 - 1.846) (0.697 - 1.256) (0.653 - 2.149)

Number of persons in the household ≥ 2 people Ref. Ref. Ref. Ref. Ref. Ref. 

 One person 1.286 0.999 1.422* 1.085 0.994 0.905

 (0.946 - 1.749) (0.634 - 1.575) (0.946 - 2.139) (0.612 - 1.924) (0.620 - 1.594) (0.430 - 1.904)

Employment status unemployed 1.501*** 1.727** 8.035*** 5.852*** 0.840 1.097

 (1.122 - 2.008) (1.134 - 2.629) (4.219 - 15.30) (2.376 - 14.41) (0.599 - 1.180) (0.635 - 1.896)

 self-employed 0.832 1.193 0.896 1.134 0.608** 0.937

 (0.626 - 1.105) (0.727 - 1.958) (0.605 - 1.327) (0.563 - 2.286) (0.402 - 0.920) (0.451 - 1.944)

 regular employee Ref. Ref. Ref. Ref. Ref. Ref. 

 non-regular employee 0.983 0.958 1.332 1.083 0.699** 0.764

 (0.774 - 1.247) (0.680 - 1.350) (0.808 - 2.196) (0.565 - 2.077) (0.527 - 0.928) (0.485 - 1.203)

Disposable income per household ≥ 6,000K Ref. Ref. Ref. Ref. Ref. Ref. 

 2,000K-< 6,000K 1.254*** 1.118 1.150 0.970 1.386*** 1.273*

 (1.075 - 1.464) (0.924 - 1.352) (0.913 - 1.448) (0.728 - 1.293) (1.126 - 1.705) (0.986 - 1.644)

 < 2,000K 1.977*** 1.441* 1.714** 1.382 1.980*** 1.464

 (1.475 - 2.650) (0.993 - 2.092) (1.082 - 2.713) (0.782 - 2.440) (1.351 - 2.901) (0.881 - 2.434)

Sleep duration of the respondent ≥ 7 hours Ref. Ref. Ref. Ref. Ref. Ref. 

 6-7 hours 1.100 1.128 0.939 0.988 1.277** 1.285**

 (0.951 - 1.272) (0.952 - 1.335) (0.752 - 1.173) (0.761 - 1.282) (1.054 - 1.547) (1.028 - 1.606)

 < 6 hours 1.644*** 1.503*** 1.613*** 1.456** 1.662*** 1.500**

 (1.355 - 1.993) (1.177 - 1.919) (1.203 - 2.161) (1.012 - 2.095) (1.283 - 2.151) (1.070 - 2.104)

Physical exercise ≥ 3 days/week Ref. Ref. Ref. Ref. Ref. Ref. 

 ≤ 2 days/week 1.175 1.066 1.128 0.979 1.265 1.184

 (0.897 - 1.538) (0.792 - 1.435) (0.768 - 1.656) (0.646 - 1.485) (0.864 - 1.851) (0.767 - 1.828)

 No exercise 1.637*** 1.418** 1.747*** 1.531** 1.540** 1.326

 (1.284 - 2.088) (1.062 - 1.895) (1.227 - 2.488) (1.013 - 2.315) (1.099 - 2.160) (0.874 - 2.012)

Smoking habit Never Ref. Ref. Ref. Ref. Ref. Ref. 

 Quit 1.135 0.929 1.119 1.054 1.176 0.856

 (0.909 - 1.419) (0.579 - 1.490) (0.794 - 1.578) (0.507 - 2.193) (0.880 - 1.570) (0.476 - 1.540)

 Sometimes + everyday 1.217 0.940 1.201 1.176 1.242 0.714

 (0.950 - 1.560) (0.520 - 1.700) (0.850 - 1.697) (0.503 - 2.750) (0.854 - 1.805) (0.302 - 1.685)

Drinking alcohol habit Never 

 ≤ 2 times/week 0.893 1.033 0.817 0.928 0.945 1.073

 (0.745 - 1.069) (0.809 - 1.320) (0.602 - 1.108) (0.609 - 1.413) (0.758 - 1.179) (0.793 - 1.452)

 ≥ 3 times/week 0.785** 0.775 0.783 0.679 0.752 0.819

 (0.629 - 0.979) (0.552 - 1.088) (0.564 - 1.087) (0.407 - 1.133) (0.547 - 1.035) (0.503 - 1.334)

 Constant 0.189*** 0.201*** 0.309*** 

 (0.129 - 0.278) (0.116 - 0.350) (0.188 - 0.506) 

 Observations 12,681 5,992 6,270 2,748 6,411 3,244

 Number of respondents 3,359 1,657 1,702 

 Hausman Test 0.0061 0.0586 0.4554

1) Adjusted odds ratios (AOR) with 95% CI (adjusted for sex, age, number of persons in the household, employment status, disposable income per household, sleep duration in weekdays, physical exercise, smoking habit, and drinking alcohol);

2) Adjusted odds ratios with 95% CI (adjusted for age, number of persons in the household, employment status, sleep duration in weekdays, physical exercise, smoking habit, and drinking alcohol);

Bold ratios: the significant results of Hausman’s test.

The levels of the General Health Questionnaire 12-item scores: 0 = ≤ 3 points, 1 = ≥ 4 points;

Robust cieform in parentheses

Ref.: Reference 1

*** p<0.01, ** p<0.05, * p<0.1

2) Response: 

Thank you for the comment. This was my careless mistake. Bold does not show ‘statistically significant results’. Bold shows the results of Hausman test. We corrected it. 

3) Response:

Thank you for finding this careless mistake. We corrected the result of the AOR for RE-female alcohol >= 3 times a week as no significant one. 

4) Response: 

The results ‘0.785** (0.629 – 0.979)’ is a significant result, as we put ‘**’. In addition, it is 0.785, not 0.7885.

5) Response: 

In estimation, the fixed-effects model uses only information of the observations that experienced changes within an individual. In other words, fixed-effects model removes the observations doesn't change over time. That is why the number of observations in FE and RE models are different. 

Discussion

14. Limitations. Dropouts are discussed in general terms. What was the actual characteristics of the dropouts in this study? The income variable is very screwed – is that sampling bias or at true reflection of the study population?

Response: 

As we wrote that ‘the dropouts might have had unhealthy outcomes or have been in unfavourable situations’, there is a possibility of sample attrition bias and dropouts. However, we explained our attrition rate in Table 1, indicating less than 10% attrition rates for 5 years. This is the same as previous studies, which revealed that the average attrition rate of panel studies revolves around 10%1). We believe that the attrition bias is not so serious, because of the low attrition rates of JHPS. 

Regarding income variable, as we noted in the reply to your comment 6, although the income distribution of our sample is not necessarily similar to the population, we confirmed that the results of weighted regression using the inverse of the number of households as weight were not so much different from the original results.　

Reference: 

1) Baltagi BH. Econometric analysis of panel data. UK: John Wiley & Sons Ltd; 2005. p. 1-9.

15. Information bias is mentioned only as recall. However, the instrument is not (or is?) validated; again, when adjusting for the number of persons in household – which is very good -the variable is not covering that, but only one or more than one. This is a serious problem and makes any conclusion related to income invalid.

Response:

As we noted in the reply to your comment 7, we confirmed that the estimation results did not change significantly even if we used the dummy variables indicating the size of household: a one-person, two-person, three-person, and four or more person dummies. Furthermore, as we noted in the replies to your comment 6, we confirmed the results of weighted regression using the inverse of the number of households as weight were not so much different from the original results. Therefore, we are afraid that this might not be a serious problem to make a conclusion about income. 

16. “Further research is needed to identify the effects of disposable income and living status on mental health conditions, as well as the role of gender”. They do exist in plenty – as do so for employment status and gender.

Response:

As you mentioned, in the next study, I would like to analyse unemployed and homemaker females separately. This would be an issue in Japanese gender studies. 

17. You do not reflect on the high prevalence of poor mental health, why? it is a central study objective. However, to examine the prevalence of poor mental health longitudinal data are not optimal, unless you want to give information on the development in mental health. Again, it is not evident if GHQ is validated in a Japanese population – and a pooled prevalence of 42% women in poor mental health does not seem reliable.

Response:

Thank you for your comment. We added the sentence into discussion as follows: 

‘The results of GHQ-12 indicated that 36.1% of men and 42.0% of women were shown to have poor mental health conditions (≥ 4 GHQ-12 score). The results indicated statistically significant differences between men and women for five years and pooled data of all waves. Hori et al. reported that 24.2% of men and 32.0% of women were shown to have GHQ score with 4 points or more (36). The prevalence of poor mental health conditions in this study was higher than those of Hori’s study. However, Matsuba et al. reported that 43.2% of Japanese people in Thailand indicated poor mental health conditions (37). According to the results of previous studies, scores for the female respondents tended to be higher than those for the male respondents. Further research is needed to determine the cause of the high prevalence rate.’

As we wrote, Matsuba et al. also reported that 43.2% of Japanese people indicated poor mental health conditions. It would be inappropriate to say that a pooled prevalence of 42% women in poor mental health does not seem reliable.

References:

1. Hori D, Tsujiguchi H, Kambayashi Y, Hamagishi T, Kitaoka M, Mitoma J, et al. The associations between lifestyles and mental health using the General Health Questionnaire 12-items are different dependently on age and sex: a population-based cross-sectional study in Kanazawa, Japan. Environ health prev med. 2016;21(6):410-21.

2. G. Matsuba, D. Suzuku, Y. Inaba. Physical practice and mental stress among Japanese people living in Thailand Juntemdo Igaku. 2007;53(4):581-7.

18. As for household income as socioeconomic index the validity is poor – first very few are in lowest category and even though it is stated the analyses are adjusted for number of persons in the household, when in fact it is only adjusting for one or more than one.

Response:

Thank you for the comment. However, we do not think that more than 350 subjects are very few as one category, while data analysis with 2 methods worked properly. As we wrote why we made three groups of disposable income for the comment 6, the categorisation method is the same as the government survey, National Survey Health and Nutrition (Ministry of Health, Labour and Welfare, Japan) in 2014. Moreover, there are previous studies in public health, which used a variable ‘household annual income’. Lei et al. (2020) included a model average household income (<1500, 1500-3000, 3000-6000, 6000-9000, <9000) 1). We also published two research papers with the variable household annual income 2, 3). As we explained several times here, the key points of our study were the relationship between whether people live alone or live with somebody and mental health conditions and impact of household annual income on mental health conditions. Please understand that as we noted in the reply to your comment 7, we confirmed that the estimation results did not change significantly even if we used the dummy variables indicating the size of household: a one-person, two-person, three-person, and four or more person dummies.

References:

1. Lei L, Huang X, Zhang S, Yang J, Yang L, Xu M. Comparison of prevalence and associated factors of anxiety and depression among people affected by versus people unaffected by quarantine during the COVID-19 epidemic in Southwestern China. Med Sci Monit. 2020;26:e924609.

2. Muto K, Yamamoto I, Nagasu M, Tanaka M, Wada K. Japanese citizens' behavioral changes and preparedness against COVID-19: An online survey during the early phase of the pandemic. PloS one. 2020;15(6):e0234292.

3. Nagasu M, Kogi K, Yamamoto I. Association of socioeconomic and lifestyle-related risk factors with mental health conditions: a cross-sectional study. BMC public health. 2019;19(1):1759.

19. As for the FE logistic regression analyses vs RE logistic analyses none of them account for the time each individual contributes with in the time series (person-years), and thus less accurate than the traditional methods used for longitudinal studies.

Response: 

Thank you for the comment. As we wrote above, the FE logistic regression analyses and the RE logistic analyses are conventional statistical methods in economics, but these have not been very common in public health yet. But Ocean et al applied fixed effect model about a longitudinal UK study on the relationship between fruit and vegetable consumption and well-being.1)

The FE logistic regression accounts for the within-person changes while the RE logistic regression does for within-person changes as well as between-person differences, so they should be more accurate than the traditional logistic regressions. 

Reference:

1) Ocean N, Howley P, Ensor J. Lettuce be happy: A longitudinal UK study on the relationship between fruit and vegetable consumption and well-being. Social science & medicine. 2019;222:335-45. 

Authors’ response to reviewers

PLOS ONE

PONE-D-20-10688

Impact of socioeconomic- and lifestyle-related risk factors on poor mental health conditions: A nationwide longitudinal 5-wave panel study in Japan

Reviewer #2: 

In general, this study is well constructed to gain a rational outcome. I have some minor comments to be addressed for better understanding of the study.

1. In the introduction section, the author mentioned that Japan was shown to be one of the countries with the highest suicide rates in the world. It is partially acceptable, but a little over-represented. Some Eastern European nations as well as Russia has higher suicide rate. Also, in latest statistics, those of the US and Sweden are not so different from Japan's. I am afraid that outdated articles the author referred can be misleading.

Response: 

Thank you for your comment. We deleted the references cited and added new data, which was provided by the OECD. The sentence was also rewritten as follows: 

‘According to data released by the OECD, the suicide rate was the top sixth in 34 countries, although the number of suicides committed by Japanese people has been declining gradually for 10 consecutive years (10).’

Reference: OECD (2020), Suicide rates (indicator). doi: 10.1787/a82f3459-en (Accessed on 03 June 2020)

2. In this study, the author classified some items the participants answered into some groups. 1) How the author decide the thresholds of each group? 2) For example, are there any reasons that people taking 6-7 hours of sleep in a day is different to those taking >7 hours sleep, not >8? In my understanding, there are rich evidence suggesting sleeping under 6 hours in a day is harmful. But how many hours you should sleep is controversial. 3) Also, the author referred a Glozier's work. But its subjects were limited to young people recommended to take 8-9 hours sleep. The author should consider to show better preceding studies.

Response: 

Thank you for the comment. For the comments 1) and 2), we have some reasons to categorise the variables into some groups. As stated in our study objectives, examining the prevalence of poor mental health conditions and to analyse the models by gender is important. We thought that analysis across gender would be more important than that using all samples simultaneously as a mental health study. Regarding the categorical variables, we divided the items into three groups to preserve sufficient samples in each group across gender as many as possible. For example, the variable about sleeping hours, this needed to be divided into 3 groups (about 30% each) for analysing the models as Table 1 shows the results. If we categorised 4 groups (≥ 8 h, 7-8 h, 6-7 h, and < 6 h), the sample size of over 8 sleeping hours was 7.2% of female subjects (n=535) and 8.7% of male subjects (n=619). This was not enough to analyse the models by gender unfortunately. 

About the average of sleeping hours among Japanese population, the Sleep Guidelines for Health Promotion 2014 published by the Ministry of Health, Labour and Welfare (MHLW) reported that about 60% of Japanese adults sleep between 6 hours and less than 8 hours and not over 8 hours; this is considered standard sleep duration. Our data showed similar trend with the report. As you recommended, analysing the association between long time sleep and mental health would be valuable. If we can publish a paper about the relationship between mental health and long sleeping hours, we will consider the sample size. 

About the comment 3), we cited one more reference and added a sentences as follows: 

‘For example, Golzier et al. analysed a cohort study and reported that shorter sleep duration is linearly associated with psychological distress in young adults (14). Lallukka et al. also reported that the combination of sleep duration and quality was associated with physical, emotional, and social functioning among Australian adults (15). ’

Table 1. Results of sleeping hours by gender. 

 Gender Total

 Female Male 

Sleeping hours ≥7 hours n 2740 2710 5450

 % 37.0% 38.2% 37.6%

 6-7 hours n 2826 2769 5595

 % 38.2% 39.1% 38.6%

 < 6 hours n 1841 1607 3448

 % 24.9% 22.7% 23.8%

Total n 7407 7086 14493

 % 100.0% 100.0% 100.0%

3. In this study, only 7.9% of the participants lived alone. According to recent official statistics in Japan, one-fourth of the household is composed of one person. In my calculation, single person household was 10.7% in 2014. Did the author compare the demographic data with those of contemporary official statistics? If there is a large discrepancy between them, the representativeness of the panel is doubtful. The same thing is also adapted to employment status, but it seems consistent with the official statistics, as far as my checking.

Response: 

Thank you for your comment. We compared the demographic data of JHPS with those of the national census taken on Oct. 2015. As you highlighted, the result of JHPS seems to have considerably fewer participants living alone than the results of the national census. As you noticed, the participants of JHPS were selected at the individual level, and those at the national census were at the household level. We also calculated the composition of single-person living at the individual level as 14.4% (the number of single person household is 18,420, which amounts to 14.4% of the Japanese population of 128,000 thousands). Therefore, there is about a double difference between population (14.4%) and our sample (7.9%). 

To account for the possible bias due to the smaller sample of single-person, we conducted a weighted logit regression using the inverse of the number of households as weight. As shown in the table below, although some of the statistical significance are different, the sign (more than 1 or not) and magnitude of the odds ratio seem not to be so different between the estimates with and without weight. Thus, we regard that although there is a sampling bias in our data in the sense that less persons living alone were selected, it does not largely affect the estimation results.

Table Results of RE and estimation with weight

　 　 Results of RE　 Results of estimation with weight 

　 　 odds ratios odds ratios

Sex Men Refe Refe 

 Women 1.419*** 1.236***

 (1.107 - 1.819) (1.122 - 1.362)

Age Under 39 Refe Refe 

 40-49 0.971 0.951

 (0.791 - 1.190) (0.866 - 1.045)

 50-59 0.836 0.861***

 (0.666 - 1.049) (0.782 - 0.948)

Number of Persons in the household ≥ 2 people Refe Refe 

 One person 1.286 1.248***

 (0.946 - 1.749) (1.091 - 1.428)

Employment status unemployed 1.501*** 1.165**

 (1.122 - 2.008) (1.022 - 1.328)

 self-employed 0.832 0.832***

 (0.626 - 1.105) (0.737 - 0.940)

 regular employee Refe Refe 

 non-regular employee 0.983 0.987

 (0.774 - 1.247) (0.887 - 1.098)

Disposable income per household ≥ 6,000 K Refe Refe 

 2,000 K-< 6,000 K 1.254*** 1.198***

 (1.075 - 1.464) (1.106 - 1.299)

 < 2,000 K 1.977*** 1.770***

 (1.475 - 2.650) (1.517 - 2.064)

Sleep duration of the respondent ≥ 7 h Refe Refe 

 6-7 h 1.100 1.028

 (0.951 - 1.272) (0.945 - 1.119)

 < 6 h 1.644*** 1.418***

 (1.355 - 1.993) (1.288 - 1.561)

Physical exercise ≥ 3 days/week Refe Refe 

 ≤ 2 days/week 1.175 1.228***

 (0.897 - 1.538) (1.054 - 1.429)

 No exercise 1.637*** 1.449***

 (1.284 - 2.088) (1.270 - 1.652)

Smoking habit Never Refe Refe 

 Quit 1.135 1.076

 (0.909 - 1.419) (0.979 - 1.183)

 Sometimes + everyday 1.217 1.126**

 (0.950 - 1.560) (1.020 - 1.243)

Drinking alcohol Never Refe Refe 

 ≤ 2 times/week 0.893 0.879***

 (0.745 - 1.069) (0.806 - 0.959)

 ≥ 3 times/week 0.785** 0.894**

 (0.629 - 0.979) (0.812 - 0.985)

 Constant 0.189*** 0.360***

 (0.129 - 0.278) (0.301 - 0.431)

 Observations 12,681 12,681

　 Number of respondents 3,359 　

Robust cieform in parentheses 

*** p<0.01, ** p<0.05, * p<0.1 

To explain the above points, we added the following sentences: 

‘First, for the proportion of single person household, we compared the demographic data of this study with those of the national census taken on Oct. 2015. The result of this study seems to have considerably fewer participants lived alone than the results of the national census. Because the participants of this study were selected at the individual level, that of the national census were at the household level. We also calculated the composition of single-person living at the individual level as 14.4% (the number of single person household is 18,420, which amounts to 14.4% of the Japanese population of 128,000 thousands). Therefore, there is about a double difference between population (14.4%) in the national census and our sample (7.9%). To account for the possible bias due to the smaller sample of single-person, we conducted a weighted logit regression using the inverse of the number of households as weight. Although there are some statistical significance between RE-CLR and RE-CLR with weight, the sign (more than 1 or not) and magnitude of the odds ratios seem not to be so different between the estimates with and without weight. Thus, we regard that although there is a sampling bias in our data in the sense that less persons living alone were selected, it does not largely affect the estimation results. ’

4. Table 2 is difficult to read at a glance because each number and the mean share the same column. The author should rearrange it.

Response: 

Thank you for your suggestions. The number of the subject and the mean are rearranged in Table 2. 

5. Is the mean income of Japanese only 542.8 JPY?

Response: 

Thank you very much for highlighting this. It was my mistake. I corrected the numbers in Table 2. 

6. What does "0 point - 12 points" mean in the GHQ score section?

Response: 

This meant that the range of GHQ scores was 0 point to 12 points. However, I deleted it. 

7. Figure 1. is hardly understandable and space-killing. What does the vertical axis mean? I reckon it shows the percentage. Anyway, it seems better to choose another type of graph (polygonal line graph or a chart maybe).　 

Response: 

Thank you for your suggestions. We tried to make another graph, but the results were not suitable to be presented as a graph. Therefore, we deleted the figure and made a table labelled as Table 2. 

Table 3. Differences in rates of GHQ scores of 4 points or more from 2014 to 2018. 

Years Females Males P value 1)

2014 41.9% 35.8% ***

2015 41.9% 35.7% ***

2016 41.8% 37.9% *

2017 43.6% 36.9% ***

2018 40.7% 34.3% **

Pooled Data 42.0% 36.1% n.s.

1) ***: p < .001, **: p < .005, *: p < .05. 

2) The levels of the General Health Questionnaire 12-item scores: Poor = ≥ 4 points, Fine = ≤ 3 points.

 

Authors’ response to reviewers

PLOS ONE

PONE-D-20-10688

Impact of socioeconomic- and lifestyle-related risk factors on poor mental health conditions: A nationwide longitudinal 5-wave panel study in Japan

Reviewer #3: 

This epidemiological survey is important for mental health field, worth reading. The authors however should 1) propose the hypothesis of the study clearly, and 2) emphasize the new findings in the present study, and 3) the differences from the previous reports throughout the manuscript.

Response: 

Thank you very much for reviewing our manuscript. We have made the suggested changes, added sentences and rewritten many parts of the manuscript.

1) We have stated our hypothesis of this study as follows: 

‘We hypothesised that better SES and healthy lifestyle-related factors may associate with better mental health conditions positively.’

2 &3) We have rewritten many parts of the abstract and added the aims clearly. The results of this study also added into the abstract:

‘This study aims to reveal the prevalence of poor mental health conditions among Japanese individuals and to identify the SES- and lifestyle-related risk factors that might lead to these conditions.’ 

‘The prevalence of poor mental health conditions, represented by a GHQ-12 score of 4 or more, was 36.1% and 42.0% of men and women, respectively.’

‘Various factors, such as unemployment, low household income, short nightly sleeping duration, and lack of exercise, showed significant longitudinal (within) associations with mental health conditions estimated by the FE-CLR models.’ 

Introduction is rewritten as follows:

‘The fixed-effects regression models are used to adjust for all time-constant unobserved confounders and decrease the risk of omitted variable bias, as well as adjust for identified time-varying confounders. ’ 

Discussion is also rewritten as follows:

‘The results of GHQ-12 indicated that 36.1% of men and 42.0% of women were shown to have poor mental health conditions (≥ 4 GHQ-12 score). The results indicated statistically significant differences between men and women for five years and pooled data of all waves. Hori et al. reported that 24.2% of men and 32.0% of women were shown to have GHQ score with 4 points or more (36). The prevalence of poor mental health conditions in this study was higher than those of Hori’s study. However, Matsuba et al. reported that 43.2% of Japanese people in Thailand indicated poor mental health conditions (37). According to the results of previous studies, scores for the female respondents tended to be higher than those for the male respondents. Further research is needed to determine the cause of the high prevalence rate.’

---

## [Decision Letter · Decision Letter 1]

23 Sep 2020

Impact of socioeconomic- and lifestyle-related risk factors on poor mental health conditions: A nationwide longitudinal 5-wave panel study in Japan.

PONE-D-20-10688R1

Dear Dr. Nagasu,

We’re pleased to inform you that your manuscript has been judged scientifically suitable for publication and will be formally accepted for publication once it meets all outstanding technical requirements.

Kind regards,

Kenji Hashimoto, PhD

Section Editor

PLOS ONE

Additional Editor Comments (optional):

Reviewers' comments:

Reviewer's Responses to Questions

**Comments to the Author**

1. If the authors have adequately addressed your comments raised in a previous round of review and you feel that this manuscript is now acceptable for publication, you may indicate that here to bypass the “Comments to the Author” section, enter your conflict of interest statement in the “Confidential to Editor” section, and submit your "Accept" recommendation.

Reviewer #2: All comments have been addressed

2. Is the manuscript technically sound, and do the data support the conclusions?

Reviewer #2: Yes

3. Has the statistical analysis been performed appropriately and rigorously? 

Reviewer #2: Yes

4. Have the authors made all data underlying the findings in their manuscript fully available?

Reviewer #2: Yes

5. Is the manuscript presented in an intelligible fashion and written in standard English?

Reviewer #2: Yes

6. Review Comments to the Author

Reviewer #2: (No Response)

7. PLOS authors have the option to publish the peer review history of their article (what does this mean?). If published, this will include your full peer review and any attached files.

Reviewer #2: **Yes: **Akihiro Shiina

---

## [Editor Report · Acceptance letter]

25 Sep 2020

PONE-D-20-10688R1 

Impact of socioeconomic- and lifestyle-related risk factors on poor mental health conditions: A nationwide longitudinal 5-wave panel study in Japan 

Dear Dr. Nagasu:

I'm pleased to inform you that your manuscript has been deemed suitable for publication in PLOS ONE. Congratulations! Your manuscript is now with our production department. 

Kind regards, 

on behalf of

Prof. Kenji Hashimoto 

Section Editor

PLOS ONE